# Electrocatalytic on-site oxygenation for transplanted cell-based-therapies

Inkyu Lee[1,15], Abhijith Surendran[2,15], Samantha Fleury [3], Ian Gimino[4], Alexander Curtiss[5], Cody Fell[3], Daniel J. Shiwarski [6], Omar Refy[7], Blaine Rothrock [8], Seonghan Jo[1], Tim Schwartzkopff[4], Abijeet Singh Mehta[2], Yingqiao Wang[1], Adam Sipe[9], Sharon John[10], Xudong Ji [2,11], Georgios Nikiforidis[2], Adam W. Feinberg [1,6], Josiah Hester[12], Douglas J. Weber [6,10,13], Omid Veiseh [3], Jonathan Rivnay [2,11,14] ✉ & Tzahi Cohen-Karni [1,6] ✉

Implantable cell therapies and tissue transplants require sufficient oxygen supply to function and are limited by a delay or lack of vascularization from the transplant host. Previous exogenous oxygenation strategies have been bulky and had limited oxygen production or regulation. Here, we show an electrocatalytic approach that enables bioelectronic control of oxygen generation in complex cellular environments to sustain engineered cell viability and therapy under hypoxic stress and at high cell densities. We find that nanostructured sputtered iridium oxide serves as an ideal catalyst for oxygen evolution reaction at neutral pH. We demonstrate that this approach exhibits a lower oxygenation onset and selective oxygen production without evolution of toxic byproducts. We show that this electrocatalytic on site oxygenator can sustain high cell loadings (>60k cells/mm$^3$) in hypoxic conditions in vitro and in vivo. Our results showcase that exogenous oxygen production devices can be readily integrated into bioelectronic platforms, enabling high cell loadings in smaller devices with broad applicability.

The transplantation of therapeutic cells, within semipermeable devices as a living pharmacy has the potential to treat a range of diseases such as endocrine disorders, autoimmune syndromes, cancers, and neurological degeneration[1–5]. Cell-based therapeutics translation to humans requires high cell densities[6] to enable miniaturized devices of therapeutic value. Studies have shown that encapsulated cells can survive at 6–10k cells/mm$^3$ densities when implanted for treating type 1 diabetes and psoriasis[7,8]. However, maintaining the potency of such densely packed therapeutic cells for extended duration is challenging due to a number of factors such as immune response from the host and inadequate availability of nutrients and dissolved oxygen[4,9]. While immunoisolation and cell protection using encapsulating size-

[1]Department of Materials Science and Engineering, Carnegie Mellon University, Pittsburgh, PA, USA. [2]Department of Biomedical Engineering, Northwestern University, Evanston, IL, USA. [3]Department of Bioengineering, Rice University, Houston, TX, USA. [4]Department of Chemical Engineering, Carnegie Mellon University, Pittsburgh, PA, USA. [5]Department of Electrical and Computer Engineering, Northwestern University, Evanston, IL, USA. [6]Department of Biomedical Engineering, Carnegie Mellon University, Pittsburgh, PA, USA. [7]Department of Physics, Carnegie Mellon University, Pittsburgh, PA, USA. [8]Department of Computer Science, Northwestern University, Evanston, IL, USA. [9]Department of Material Science and Engineering, The Pennsylvania State University, State College, PA, USA. [10]Neuroscience Institute, Carnegie Mellon University, Pittsburgh, PA, USA. [11]Simpson Querrey Institute, Northwestern University, Chicago, IL, USA. [12]Interactive Computing and Computer Science, Georgia Institute of Technology, Atlanta, GA, USA. [13]Department of Mechanical Engineering, Carnegie Mellon University, Pittsburgh, PA, USA. [14]Department of Materials Science and Engineering, Northwestern University, Evanston, IL 60208, USA. [15]These authors contributed equally: Inkyu Lee, Abhijith Surendran. ✉e-mail: jrivnay@northwestern.edu; tzahi@andrew.cmu.edu

selective membranes or engineered biomaterials can partially address immunoreactivity, nutrient and oxygen insufficiency presents a critical challenge. Oxygen has been regarded as the limiting factor supporting cell viability and potency[10]. Due to the oxygen mass diffusion limit, in a native tissue each cell is within ca. 100 μm from a blood capillary to allow adequate oxygen supply[10]. Transplanted exogenous cells or tissue require the formation of new blood vessels or supplemental oxygenation[11]. Delay in vascularization limits the success of the transplantation as well as the transplanted cells' function. Oxygen deficiency in the transplanted cells is caused by (1) insufficient oxygen tension at implantation site, (2) innate large oxygen consumption of cells (i.e., metabolic demand), (3) high cell density, (4) additional barriers to oxygen diffusion (i.e., membranes, or formation of encapsulating fibrotic tissue).

To address the hypoxic stress on transplanted cells, various strategies have been investigated to enhance exogenous oxygen delivery. Two major strategies employed to mitigate oxygen deficiency may be classified into active and passive methods. Active methods involve oxygen release through an externally controllable mechanism, e.g., delivery of gaseous oxygen to the transplanted cells (i.e., islet cells)[12]. Passive methods rely on gradual release of oxygen through unregulated or self-regulated mechanisms, e.g., engineered platforms to increase the oxygen exchange with the implantation environment[13] or release of oxygen from metal peroxides[14–16]. Though these approaches are able to support transplanted cells they are limited in control of oxygen release, lifetime of available oxygen supply and limited supported cells' density, e.g., pancreatic islet (10k cells/mm$^3$)[7] and engineered cells (6k cells/mm$^3$)[8].

Electrochemical water electrolysis for oxygen production is a promising approach for providing oxygen to cells[17,18]. However, its demonstration in vivo has been limited due to improper materials selection geared toward efficient water splitting, the use of bulky and complex electronics for splitting water and limited power budget. Although electrocatalytic water splitting is widely accepted in renewable energy research, e.g., fuel cells, its application in tissue engineering has been limited due to the sluggish nature of water splitting in neutral environments[19]. The pH-dependent nature of water's redox reactions and its inherent thermodynamic stability further hinder electrochemical water decomposition in physiological environments, as they lack the necessary active species like H$^+$ and OH$^{-}$[20]. Additionally, neutral pH necessitates a higher overpotential, leading to increased power consumption and the potential risk of generating toxic byproducts such as chlorine through chloride oxidation ($E_{OX}$(Cl$^-$) = 1.771 V vs. RHE). Consequently, it is crucial to employ electrocatalysts that can effectively lower the energetic barrier while being biocompatible and highly selective for oxygen evolution reaction (OER) at neutral pH.

Iridium oxide is a benchmark electrocatalyst for OER due to its exceptional catalytic efficiency and stability[21,22]. Unlike other electrocatalytic materials, iridium oxide retains its robust catalytic activity in acidic environments where water molecules are directly involved in the OER[19,22–25]. This makes it suitable as an electrocatalyst in physiological environments, as the electrochemistry of OER is fundamentally similar in neutral and acidic conditions[20]. Its biocompatibility has been validated by extensive research outcomes in the electrophysiology community[26–30]. Iridium oxide films can be produced with nanoscopic morphologies, which allow device miniaturization and enlarged electrochemical surface area readily available for electrocatalysis[27,29,30].

Here we report a highly controlled, on demand electrocatalytic on-site oxygenator (ecO$_2$) platform designed to safely support implanted therapeutic cells. Capitalizing on its unique catalytic properties, stability, patternability and biocompatibility, we employed a nanostructured sputtered iridium oxide film (SIROF) to enhance the kinetics of OER and reduce the energetic cost to produce oxygen. Detailed engineering of the ecO$_2$ geometry resulted with precise control over the distribution of generated oxygen. In vitro

experiments assessed the platform's ability to sustain cell viability while maintaining therapeutic peptide secretion in high-cell density (60k cells/mm$^3$) under hypoxic conditions (1% O$_2$) up-to 3 weeks. ecO$_2$ effectiveness was subcutaneously tested in vivo (rat) for 10 days, showing its promise as a tool for cell-based therapeutic interventions, and paving the way for biomedical engineering applications, e.g., transplanted cells to treat diabetes[31] or cancer[32].

## Results

### De novo design of ecO$_2$

The ecO$_2$ platform comprises a specialized chamber for housing high-density therapeutic cells and an integrated oxygenation system to provide oxygen support, implanted at readily accessible and clinically relevant locations such as subcutaneous and intraperitoneal sites (Fig. 1a). Oxygen generation is achieved through electrolysis of water and precisely regulated using a battery powered and wirelessly controllable electronic system, although a fully battery-free wireless power technology can be readily implemented with ecO$_2$. As a key material for oxygen production, sputtered iridium oxide (SIROF), which is a highly active and biocompatible electrocatalyst for water oxidation, was adopted in our system. To ensure uniform oxygen delivery toward implanted cells, we designed various array geometries and estimated the distribution of the generated oxygen from them, guided by finite element analysis (FEA) (Fig. 1b). Given the symmetry of the designed electrode, 2-dimensional oxygen profiles were studied for cross-section of the electrode (Supplementary Fig. 1). Based on simulation studies demonstrating a uniform oxygen distribution profile, we utilized an array composed of point sources (Fig. 1c). The fabricated electrode configuration consisted of a central SIROF anode and a surrounding platinum cathode, resulting in highly distributed oxygen profiles. Fabrication methods and procedures are illustrated in "Methods" and Supplementary Fig. 2. Whereas SIROF anode possesses smaller geometric surface area in comparison to Pt cathode, nanostructured SIROF enhances catalytic activities of iridium oxide toward oxygen evolution reaction, thus contributing to an enlarged electrochemical surface area. We maximized the surface area of the cathode where hydrogen is evolved to minimize current density and prevent bubble formation resulting from the cathodic reaction, which occurs due to the inferior solubility of hydrogen in water[18]. The synthesized catalysts for ecO$_2$ exhibited nanoporous structure, which is advantageous for triggering electrocatalytic oxygen evolution reaction at lower potential (Fig. 1d), thereby reducing power requirements for the electronics as well as preventing electrochemical side reactions which can occur upon high potential application. Surface analysis evidenced the presence of rutile IrO$_2$ interspersed with amorphous regions (Supplementary Fig. 3), as well as the existence of electrocatalytically active Ir$^{4+}$ and oxygen vacancies for OER (Supplementary Tables 3 and 4 and Supplementary Fig. 4). The combination of highly active amorphous iridium oxide and stable crystalline IrO$_2$ enabled the support of implantable cell therapies through chronic oxygen delivery via electrochemical water splitting[33], recapitulating the critical features necessary for on-demand oxygen production.

### Effective, stable and safe on-site oxygen delivery

We benchmarked the activity and stability of the SIROF-based catalyst electrode against platinum electrode in 1× PBS electrolyte (pH=7.4). Linear sweep voltammetry (LSV) scan in anodic direction revealed the characteristic activation signal of iridium oxide in OER at approximately 1.6 V vs. RHE[34,35]. The SIROF catalyst demonstrated significantly lower overpotential of 395 ± 18 mV compared to Pt electrode's 565 ± 10 mV (Fig. 1a) representing reduced energetic cost to produce oxygen with catalytic activity of SIROF. Higher current observed for SIROF at the same applied potential implied its ability to achieve power-efficient electrocatalytic oxygen generation in bioelectronic applications. The stability of SIROF was examined under continuous

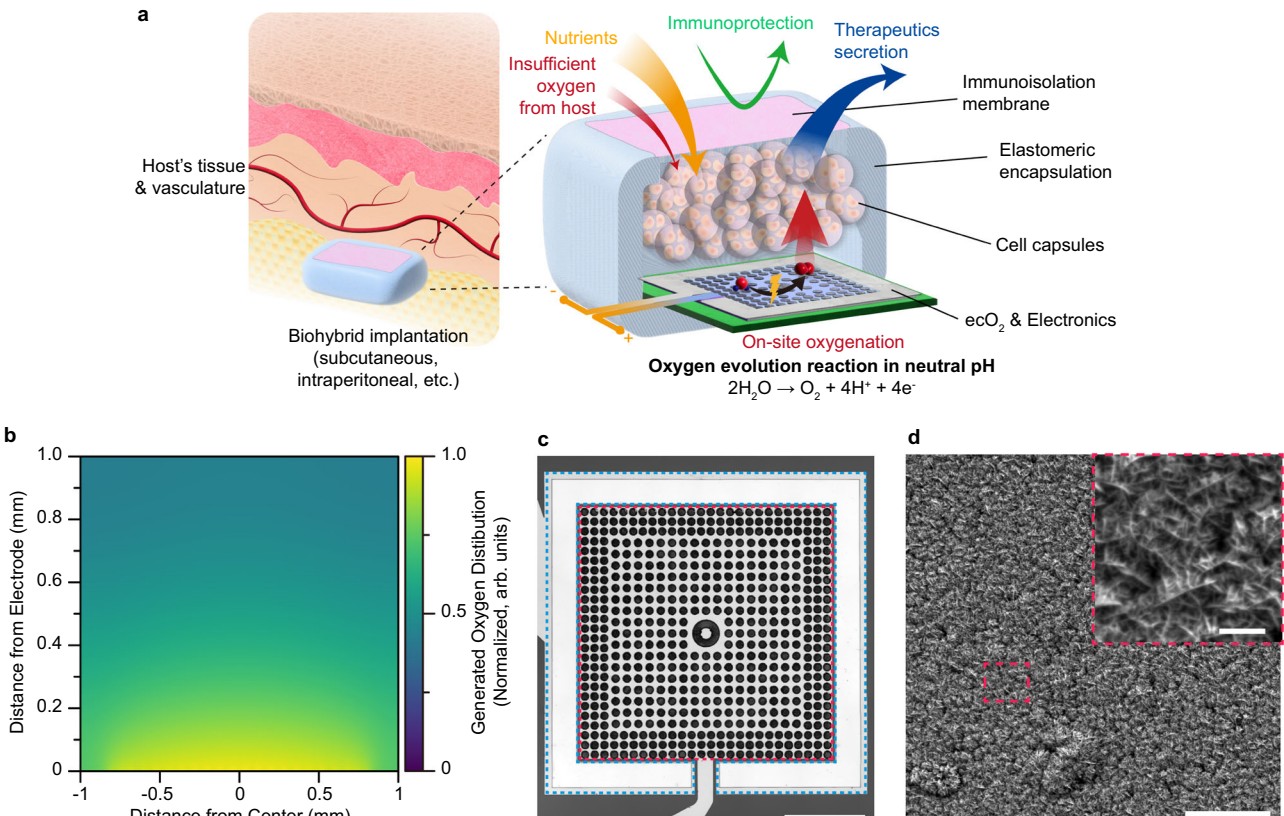

**Fig. 1 | Electrocatalytic on-site oxygenation (ecO2) for implantable cell therapies platform. a** A schematic illustration of ecO$_2$; the image is an original creation of the authors. **b** A representative optical image of fabricated ecO$_2$ device; blue dashed outline: Pt counter electrode; red dashed outline: working electrode with micropatterned circular sputtered iridium oxide (SIROF) catalytic array; scale bar: 500 μm. **c** Oxygen distribution profiles from the designed catalytic arrays. Oxygen levels were normalized with the value at the bottom of the chamber ($y = 0$ mm). **d** A representative scanning electron microscopy (SEM) image of SIROF; scale bar: 5 μm; inset: 500 nm; imaging was repeated in 10 random spots.

chronoamperometric operation with a 2-electrode setup at 1.7 V over 14 days. To monitor the change in catalytic activities, LSV curves in 3-electrode and 2-electrode setup were collected every 24 h. The onset of oxygen evolution for SIROF remained largely unchanged during the 14-day experiment, indicating that the applied catalyst did not experience significant degradation under continuous electrical stress (Fig. 2b). Additionally, the negligible onset change observed in 2-electrode settings indicated the overall stability of ecO$_2$ over 14 days of oxygen evolution (Supplementary Fig. 5).

ecO$_2$ can evolve oxygen in a highly controlled manner, without producing deleterious side products. Oxygen concentration was measured during potential pulsing (duration: 30 s). ecO$_2$ was able to produce oxygen at levels exceeding atmospheric levels (21%, 8.238 mg L$^{-1}$ at 25 °C) when applying potentials greater than 1.6 V. Furthermore, bioelectronic control allows for modulation of oxygen tensions in a precise manner, by applying different potentials (Supplementary Fig. 6a, b) or changing the duty cycling (Supplementary Fig. 6c, d). Lower duty cycles provide an added benefit of extended device lifetime by curtailing power consumption and electrochemical stress on the catalytic arrays. While oxygen generation is validated, selectivity of oxygen production must also be confirmed in complex physiological environments. To this end, we monitored possible side-reactions such as chloride oxidation, peroxide formation, and pH change during 24-hour continuous oxygen generation. While there is a possibility of generating other reactive oxygen species (ROS) such as hydroperoxide ion, superoxide ion, hydroxyl radical, and singlet radical, these ROS are not a major concern in the context of the current study. These are unstable at neutral pH and undergo disproportionation or dismutation reactions and end up as hydrogen peroxide in the

electrolyte[36–38]. Since OER in neutral pH results in protons as a byproduct ($2H_2O \rightarrow O_2 + 4H^+ + 4e^-$), local pH change may occur. However, pH was maintained within the buffer capacity of 1× PBS, which adequately mimics the buffering capacity of body fluids[39]. Measurements of evolved hydrogen peroxide showed detectable levels at potentials above 1.9 V. Even so, the concentration of generated hydrogen peroxide remained below the intracellular level of ca. 100 nM up to 2.2 V[40]. Finally, chlorine, which comes from chloride oxidation ($2Cl^- \rightarrow Cl_2 + 2e^-$), was not detected below 1.9 V. Since chloride is one of the most abundant ions in biofluids, its oxidation was the most concerning side reaction. These combined results establish a wide operation window for safe and selective oxygen evolution of ~300 mV, where OER onset occurs at 1.6 V, and harmful side reactions begin at 1.9 V (Fig. 3c). This window may be further expanded to higher voltages with the implementation of a selective coating or membrane that blocks reactants (i.e. Cl$^-$ ions) from reaching the working electrode. It is this window of operation that enables the highly tunable on-site production of oxygen to support high cell densities of non-native cells.

## ecO$_2$ safely sustains high cell density in vitro

Supplemental oxygenation is required to sustain the metabolic needs of cells when packed at high densities. When there is an imbalance of O$_2$ availability to sustain dense cell needs, encapsulated cells undergo apoptosis[7]. To test this hypothesis, we loaded ARPE-19 cells in spherical capsules (diameter of 415 ± 31 μm) at 60k cells/mm$^3$ density and exposed cultured in vitro for 21 days with and without O$_2$ (For additional information see Materials and Methods). ARPE-19 cells were chosen as a model chassis for investigation because of their clinical

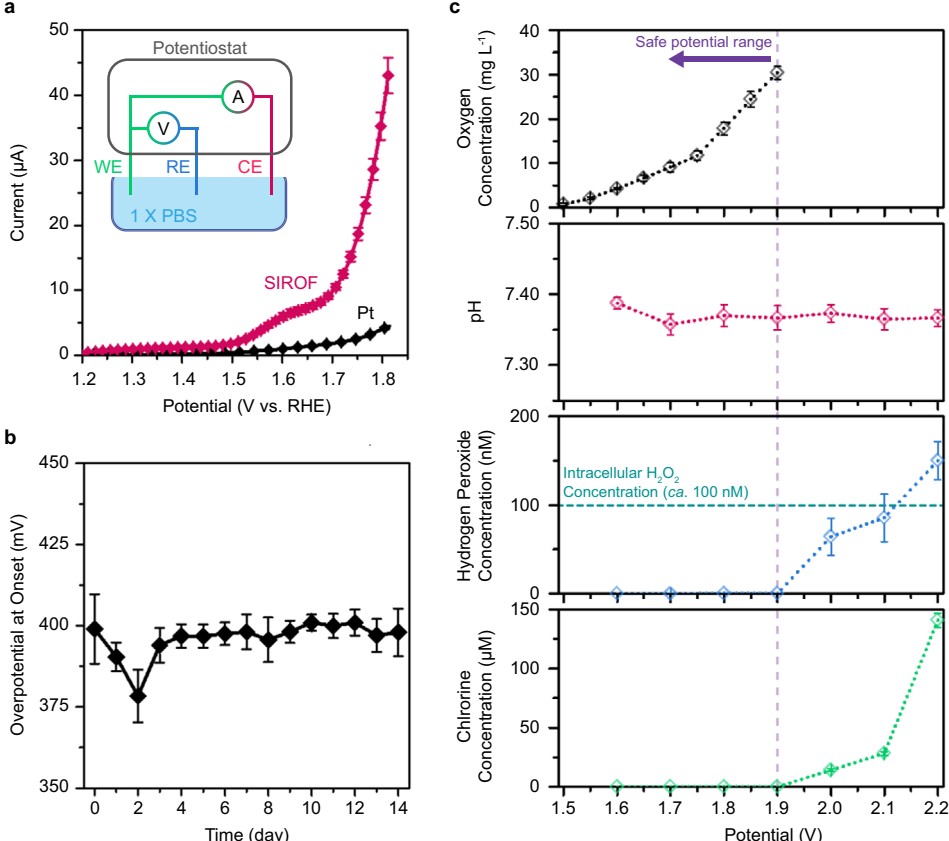

**Fig. 2 | ecO₂ is an effective, stable and safe oxygen generation platform. a** Linear sweep voltammetry (LSV) measurement in anodic direction of pristine platinum (Pt) (*n* = 8 independent electrodes, gray) and sputtered iridium oxide (SIROF) electrocatalysts (*n* = 8 independent electrodes, red); inset−an illustration of the 3-electrode electrochemical characterization setup; WE working electrode, catalyst array, RE reference electrode, 1 M Ag/AgCl, CE counter electrode, Pt mesh. **b** The stability of SIROF as an electrocatalyst, presented with overpotential change over 14 days at onset (3-electrode setup, *n* = 5 independent electrodes). **c** Oxygen generation at different applied potential and its side reactions (2-electrode setup, *n* = 4 independent electrodes); oxygen concentration−black; pH−red; hydrogen peroxide−blue; chlorine−green. Oxygen was produced with 30-s pulse duration at each potential, while side reactions were detected after 24-h chronoamperometry at each potential (*n* = 4 independent devices). All data presented with mean ± SD.

relevance to a wide range of cell-based therapeutic products for oncology[3,32] (NCT05538624), eye disorders (NCT04577300), Blood disorders (NCT04541628), enzyme replacements (NCT05665036), and Alzheimer's[41]. This cell line is widely used in the clinic because it is non-tumorigenic[42], displays contact-inhibited growth characteristics[43], amenable to genetic modification[42], and in human trials, it has been shown to be safe[41,44,45]. By using a non-dividing cell line, we are able to assess potency losses because of cell death reliability owing to the fact that in our device, the transplanted ARPE-19 cells would not be mitotically active. By contrast, previous studies[4,46] with encapsulated cell therapy devices focused on non-clinically translatable cancer cell lines, i.e., HEK 293 cell chassis, which have the mitotic capacity in vivo and new cell replication can compensate for cell death from hypoxia.

High density cell clusters encapsulated in alginate capsules were adopted as an in vitro model to simulate high cell density transplantation. Approximately 100 capsules were added in each catalytic array. While nutrients other than oxygen were supported by periodic media exchange, it was assumed that oxygen supplementation was exclusively dependent on ecO₂ oxygen production, because all cell culture media was deoxygenated under 1% oxygen. Evaluated at different time points using fluorescence confocal microscopy live/dead assay of the oxygenated cells showed improved cell viability of 83.1 ± 7.5% (Fig. 3a) after 21-day incubation in hypoxia, while hypoxic incubation without oxygenation (negative control) showed massive cell death (viability: 8.6 ± 5.6%) (Fig. 3b). The obvious contrast in 3D-rendered cell capsule images confirmed that oxygen is indeed the limiting factor of high cell

densities. While progressive decline of cell viability in negative control was attributed to hypoxic stress, only 7.5% of live cells were dead after 3-week culture compared to the initial stage of culture (89.8 ± 2.8% at day 0) (Fig. 3c). Notably, SIROF catalysts that underwent a continuous 21 days 100% duty cycle exhibited nanoporous morphology with apparent degradation of the dendritic features (Supplementary Fig. 7). This degradation was also observed in SIROF used in continuous stimulation of neural electrodes[45]. Interestingly, such significant oxygen demand in highly dense cell capsules led to hypoxic cell death even under normal oxygen level (20%, positive control) (Supplementary Figs. 14 and 15), confirming that on-site oxygen production beyond atmospheric oxygen level is crucial to address oxygen deficiency for dense or large cell constructs. Cell viability distribution along the z-axis showed no hypoxic core formation for the cell-loaded capsules that were oxygenated with ecO₂, representing negligible variation between the edges and core. This insignificant difference implied that the cell death in these samples was not due to limitation in oxygen diffusion. In contrast, the samples without ecO₂ showed a clear cell death predominantly away from the surface, evidencing that diffusion limitation caused cell death with the formation of hypoxic core (Fig. 3d, e and Supplementary Fig. 16).

While the viability of cells was improved by electrochemical oxygenation, the production of a peptide (or other transcriptional biomolecule) directly indicates maintenance of cellular potency toward applications in on-site therapy production. In fact, cell functionality is more sensitive to oxygen tensions because the dysfunction

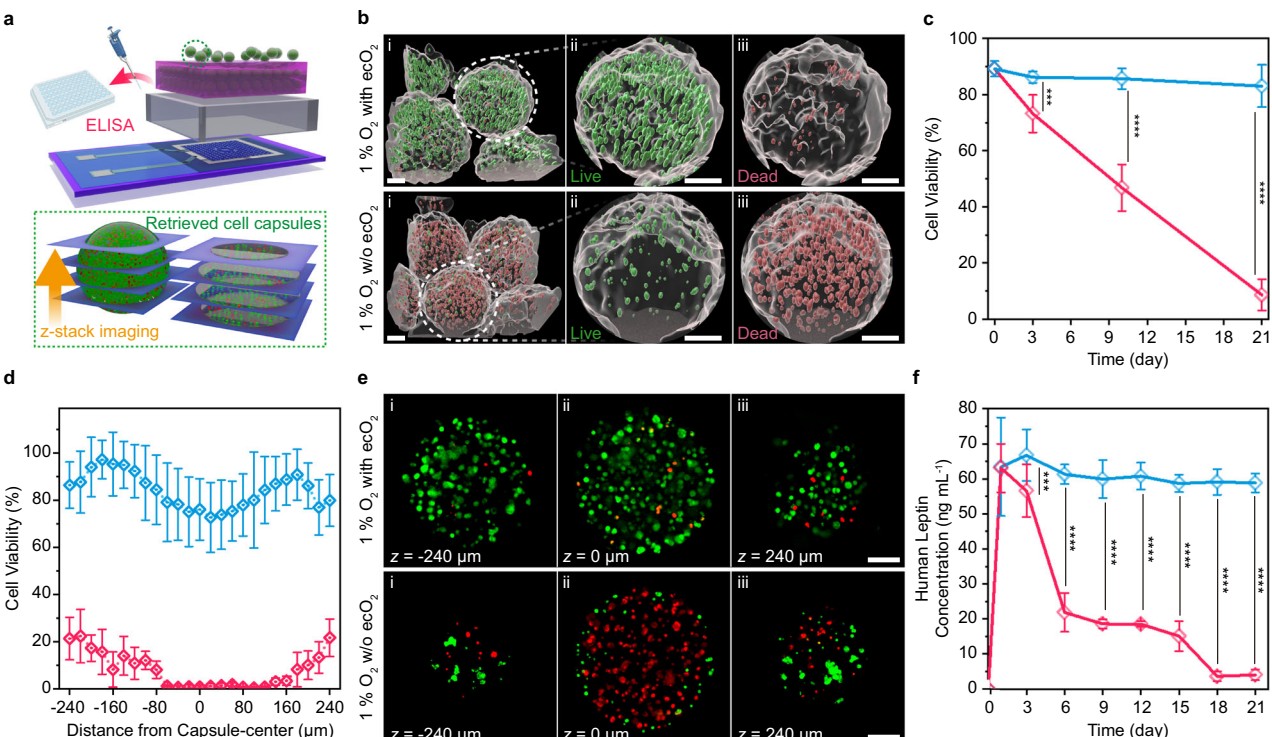

**Fig. 3 | ecO$_2$ supports cell capsules in vitro for 21 days. a** A schematic illustration of the in vitro oxygenation assay and analysis; the image is an original creation of the authors. **b** A representative set of 3D-rendered live/dead assay fluorescence images after 21-day in vitro oxygenation; i−a representative 3D-rendered z-stacked images, ii−expanded view of the marked white dashed circle, live cells, iii−dead cells. All scale bars are 100 μm. **c** Cellular viability as a function of time ($n = 4$ biologically independent samples); blue−1% oxygen with oxygenation; red−1% oxygen without oxygenation; results are presented with mean ± SD; ***$p < 0.001$, ****$p < 0.0001$; $p_{day\ 3} = 9.84 \times 10^{-4}$; $p_{day\ 10} = 8.52 \times 10^{-27}$; $p_{day\ 21} = 1.39 \times 10^{-46}$. **d** Single-capsule-level spatial distribution of cell viability in cell laden alginate capsules ($n = 158$ capsules for oxygenated cells (blue) and $n = 122$ capsules for 1% oxygen without oxygenation (red)); results are presented with mean ± SD. **e** A

representative live/dead assay at single-capsule-level of 21-day hypoxic incubation with or without oxygenation; i−$z = -240$ μm; ii−$z = 0$ μm; iii−$z = 240$ μm; scale bars are 100 μm; imaging was repeated in 25 and 20 times for the samples with or without oxygenation, respectively. **f** Peptide production as a function of in vitro culturing time. Blue corresponds to 1% oxygen with ecO$_2$ (oxygenation in hypoxia, experiment, $n = 4$ biologically independent samples); red corresponds to 1% oxygen without ecO$_2$ (hypoxic incubator, negative control, $n = 4$ biologically independent samples); ***$p < 0.001$; ****$p < 0.0001$; $p_{day\ 3} = 1.32 \times 10^{-4}$; $p_{day\ 6} = 5.32 \times 10^{-12}$; $p_{day\ 9} = 3.23 \times 10^{-29}$; $p_{day\ 12} = 2.93 \times 10^{-36}$; $p_{day\ 15} = 1.75 \times 10^{-15}$; $p_{day\ 18} = 3.92 \times 10^{-27}$; $p_{day\ 21} = 6.36 \times 10^{-28}$; results are presented with mean ± SD. One-way ANOVA and post-Tukey analysis was performed for one-sided mean comparison.

emerged at higher levels of oxygen than hypoxia-induced cell death[47]. The ARPE-19 cells herein were engineered to produce and secrete leptin, a therapeutic hormone[48] used for applications in obesity, endocrine disorders, and regulation of circadian rhythm. Leptin levels in vitro were quantified via human leptin enzyme-linked immunosorbent assay (ELISA) over a 24 h production period. Noticeably, the concentration of secreted leptin from the engineered ARPE-19 cells represented a similar tendency to viability, as depicted in Fig. 3f. While the amount of the produced leptin declined over time in negative control, cells in the ecO$_2$ oxygenation condition showed consistent levels of leptin production and release over the 3 weeks in vitro at high cell loadings.

## ecO$_2$ supports implanted cells in vivo

Oxygen tension changes in different tissues, e.g., ca. 50, 40 and 30 mmHg O$_2$ in the kidney, pancreas, and general intraperitoneal space respectively[49]. To test the efficacy of the ecO$_2$ in vivo we chose to implant the platform subcutaneously where the oxygen tension is lower ($39.0 \pm 6.3$ mmHg O$_2$). We demonstrated oxygen delivery to implanted cells in a rodent model (for additional information, see "Methods") for 10 days, as we observed that 10-day hypoxic incubation gave rise to the significant difference in cell viability. While medical grade PDMS served as impermeable housing for electronics, a microporous polycarbonate membrane was added at the top of the cell compartment to provide selective mass transport for nutrients other

than oxygen (Supplementary Figs. 21 and 23)[4]. The implanted ecO$_2$ consisted of a cell capsule compartment supported by 8 catalytic arrays for ca. 800 alginate cell capsules (60k cells/mm³). The ecO$_2$ was implanted at the abdominal skin pocket and operated by a remote-controllable potentiostat circuit secured on rat's back which was connected to the cell compartment via a flexible tether (Fig. 4a and Supplementary Figs. 21 and 22). Note that the study did not distinguish between the sexes of the animals, and multiple samples were used for each sex. The devices were retrieved and analyzed to evaluate cell viability after 10 days. Based on the visual inspection upon explantation of the devices, there was no apparent indication of fibrosis or fluid accumulation from inflammation on either device or the implanting locations (Supplementary Fig. 23). Capsules with ecO$_2$ support showed significantly higher cell viability (73.7%) than capsules without ecO$_2$ support (26.6%) (Fig. 4b). The relatively lower viability compared to 10-day in vitro oxygenation implies that the volume of media per cell (in vitro: 2.5 μL per capsule, in vivo: 0.27 μL per capsule) and media exchange may affect the viability of the cells. Nonetheless, as depicted in Fig. 4c, significantly improved viability of the oxygenated cells highlighted the effectiveness of ecO$_2$ to support implantable cell therapies.

## Discussion

On-site electrocatalytic production of oxygenation demonstrates maintenance of cell viability and therapy production at high cell

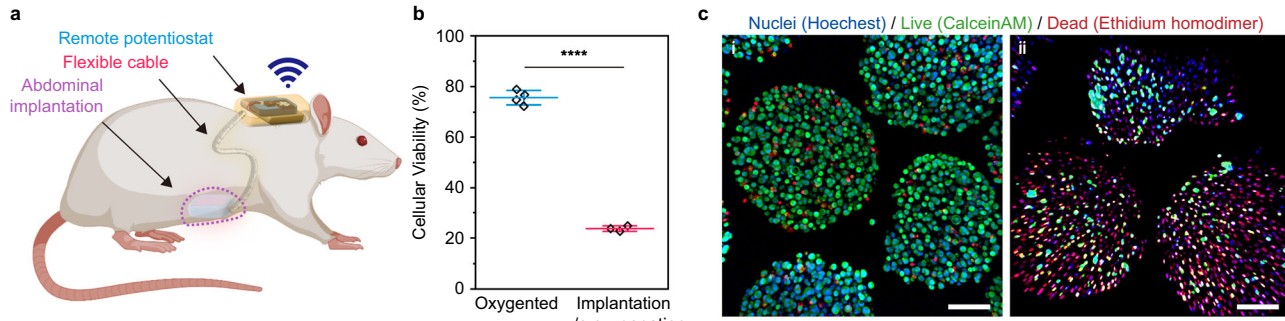

**Fig. 4 | ecO₂ maintains cellular viability in vivo. a** A schematic illustration of the in vivo oxygenation; an image of a rat was taken from Biorender.com and recreated by the authors under the agreement ZP25ZLTQNP. **b** The cell viability after 10-day implantation in rat with (blue) or without oxygenation (red); results are presented as mean ± SD; bar−mean; whisker with caps−standard deviation (*n* = 4 animals with oxygenation and n = 3 animals without oxygenation); Student t-test was performed for one-sided mean comparison; \*\*\*\**p* = 2.49×10⁻⁵⁰. **c** A representative set of live/dead assay fluorescence MIP images after 10-day in vivo oxygenation; i−implantation with oxygenation; ii−implantation without oxygenation; scale bar: 100 μm; imaging was repeated in 25 random spots each devices from independent animals.

loadings, enabling more compact cell therapy devices with high dose potential. This is achieved by supplying physiologically required levels of dissolved oxygen in a highly controlled manner (through bioelectronic control of current or voltage), without formation of gas bubbles or generation of hyperoxic (high oxygen) conditions often showing toxic effects. Importantly, this is possible due to the wide operation window for selective oxygen evolution: limiting the side reactions that arise from electrochemistry of species present in the complex cellular environment. In this work, this concept was demonstrated with ARPE-19 at loadings of 60,000 cells/mm³ (ca. 800 capsules) in vitro and in vivo, using proper selection of electrocatalyst composition and surface morphology, namely SIROF.

Notably, this work develops the components for oxygenation, which can be readily integrated with other developed technologies for sensing, actuation, communication, and power transfer/generation in an implantable device. For example, while we use batteries in the implant in this work to ease experimentation, the low power requirements (ca. 1.25 μW) (especially at low duty cycling) can be readily integrated with wireless power/communications technologies with more favorable safety profiles (i.e., radio frequency/ultrasound/magnetoelectric[50,51]). The number of devices in ecO₂ can be varied depending on the specific requirements of the therapy, such as the desired cell volume, total cell number, or needed oxygenated area. Furthermore, ecO₂ can enable feedback-controlled oxygenation by directly measuring O₂ concentrations or other metabolic markers to maintain a pre-defined oxygenation setpoint. While this work sets the groundwork for a readily integrated and simplified oxygenation system, future work should address the longevity of the system, exploring non-thin film form factors, especially for application in tissue integration, and expanding the operating winder and selectivity of OER production through the incorporation is novel and existing biomaterials and coatings, including anti-fouling layers, and selective membranes.

This platform for on site, local, and regulated oxygen production suggests broad applicability in biomedical settings. In the cell therapy space, noted herein, ecO₂ can benefit islet transplantation, as well as cell therapies for onco-therapies, treating metabolic disorders, pain, addiction, and others. Furthermore, given the 14-day period needed to vascularize implanted tissues and cell therapies[52], temporary oxygenation of tissue transplants can bridge the time from implantation to integration/vascularization from the host recipient. Such an oxygenation platform could be integrated into transport chambers and carriers for long distance transport or donor cells, tissues, or organs. Finally, this device concept can be integrated as a research tool to stress oxygen tension in in vitro organoid models for basic research. As such, a controlled oxygenator that is readily integrated with bioelectronic

and biomaterial platforms presents breakthrough opportunities for therapeutics and regenerative engineering applications.

## Methods

All procedures were approved by the Carnegie Mellon University Institutional Animal Care and Use Committee and the DoD Animal Care and Use Review Office (IPROTO202100000285).

### Finite element analysis for oxygen diffusion simulation

The COMSOL simulation was constructed in 2-dimensional geometry, which assumed that cross-sectional diffusional profiles were uniform along the axis normal to the plane displayed because of symmetric natures of the designed arrays. The simulations were carried out in a 1 mm high × 2 mm wide rectangle. Oxygen-evolution electrodes were modeled as flat surfaces with pattern spacings according to the electrode designs. The applied physics module, Transport of Diluted Species, limited the oxygen fraction up to their solubility limitation and Fick's Law was selected as a standard diffusion mechanism with the diffusion coefficient of $2×10^{-9}$ m² s⁻¹ for dissolved oxygen in water. The resulting profiles were linearly correlated with applied oxygen flux, the distribution of oxygen was normalized with the oxygen level at the final time stamp.

### ecO₂ catalytic array fabrication

Prior to the sputtering of metals, Si/SiO₂ (600 nm) wafers were cleaned with sonication in acetone for 10 min, rinsed with IPA and N₂ blow dried. The wafers were treated in a barrel etcher with O₂ plasma at 100 W for 2 min, immediately followed by spin-coating of LOR10B (MicroChem) at 3000 rpm for 40 s and baking it at 190 °C for 5 min. After applying photoresist (Shipley, Microposit S1813) at 3000 rpm for 40 s and baking it at 115 °C for 5 min, UV exposure for photopatterning was carried out with a 4-inch mask in MA6 mask aligner (Karl Suss MA6) for 100 s. Post-exposure, patterns were developed in 2.6% tetramethylammonium hydroxide aqueous solution (Shipley, Microposit MF-CD-26 Developer) for 23 s. The patterned wafers were metalized with a layer of Ti (250 nm) and Pt (450 nm) using DC sputtering at 200 W. Lift-off was conducted in Remover PG (Kayaku Advanced Materials, Inc.) for 30 min at 65 °C. On the metalized wafer, the catalytic array (anode, working electrode) was patterned with Ti (150 nm) and Ir (250 nm) via DC and RF sputtering at 200 W, respectively, while the cathode (counter electrode) remained as intact Pt surface. Reactive sputtering of SIROF was immediately performed after the Ir layer deposition. SIROF was synthesized via reactive sputtering with oxygen plasma[53,54]. Sputtering was performed in a house-built sputtering system at RF power of 300 W at 10 mTorr with 1:1 mixture of argon and oxygen. A 7 min sputtering resulted in 1.5 μm thick SIROF on an

iridium-topped metal stack (Ti 250 nm/Pt 450 nm/Ir 150 nm), which enhances the adhesive properties of SIROF to the metals. Excessive materials were lifted off in Remover PG (MicroChem) for 30 min at 65 °C. The patterned arrays were passivated with polymeric window passivation (Kayaku Advanced Materials, Inc, SU-8 3010), followed by hard baking at 180 °C for 30 min.

## Raman Spectroscopy

Raman spectra of SIROF were collected using NT-MDT Spectra with 532-nm excitation through 100x/0.7 NA objective. The spectra were recorded with a 0.5 neutral density (ND) filter and 30-second acquisition time, acquired from 10 random locations of 4 independent samples. While detected $E_g$ (ca. 550 cm$^{-1}$), $B_{2g}$ and $A_{1g}$ (ca. 720 cm$^{-1}$) modes indicate crystalline synthesized SIROF, overwrapped $B_{2g}$ and $A_{1g}$ peaks showed amorphous domains also existed in the materials.[55–57] In addition to main peaks, weak mode at ~365 cm$^{-1}$ also supported the presence of amorphous iridium oxide, derived from stretches of disordered iridium oxide structure[58].

## Grazing incidence X-ray diffractometry (GIXRD)

To investigate crystallographic features of SIROF, we acquired the XRD spectra using grazing incidence to avoid interference from materials stacked under the films. The GIXRD patterns were collected using Malvern Panalytical Empyrean XRD with a wavelength of 0.154 nm (Cu Kα) and a grazing angle of 3° at 45 kV and 40 mA. The scan step size was 0.02° with 1 s per scan. The diffraction patterns at 28.314°, 34.716° and 40.350° are corresponding to (110), (101) and (200) plane of rutile IrO$_2$ lattice, respectively, which implied that dendritic SIROF nanostructures were composed of nanocrystals[34,57,58].

## X-ray photoemission spectroscopy (XPS)

SIROF surface chemical composition was investigated using XPS. The spectra were acquired using Thermo Scientific ESCALAB 250Xi equipped with a monochromatic KR Al X-ray source (spot size: 500 μm). Survey spectra were collected with a 120-eV pass energy, 15-ms dwell time and 1-eV step size. High-resolution scans were performed in Ir 4 f, O 1 s and C 1 s regions with 0.1-eV step size, and C-C peak was utilized for calibration as 284.8 eV. The trace level of Si signals implied that collected XPS spectra were free from the substrate (Si/SiO$_2$ (600 nm thermal oxide) wafer). The recorded XPS signals were analyzed using Avantage (Thermo Scientific) software. Ir 4 f doublet peaks were deconvoluted using asymmetric curve fitting with Ir (IV) and its first satellite signal[59]. Corresponding to electrochemical analysis displaying the absence of the transition from Ir (III) to Ir (IV), iridium species in SIROF represented +4 oxidation state, which is active for the electrocatalysis of oxygen evolution reaction[35]. Note that including Ir(III) species did not create reasonable results, which indicated +4 oxidation state is the only chemical status that existed in SIROF. A high-resolution scan of O 1 s regime indicated the synthesized SIROF catalysts implied that synthesized SIROF possessed oxygen vacancies as well was lattice oxygen[60,61]. Because while oxygen vacancy signals are originated from amorphous iridium oxide (IrO$_X$), lattice oxygen is detected in IrO$_2$ crystal, the chemical status of oxygen in SIROF provides a hint of semi-crystalline characteristics. Combined with the structural analyses showing both amorphous characteristics (Raman spectra) and crystalline (GIXRD), the mixed signals from lattice and vacancy oxygen in XPS analysis substantiated that SIROF has semi-crystalline natures. As reported that oxygen vacancies are key components in oxygen evolution mechanism of iridium oxide catalysts[33], the existence of Ir (IV) states and oxygen vacancies indicated that synthesized SIROF has catalytic activity in OER.

## Scanning electron microscopy (SEM)

SEM images were acquired by using a field emission gun SEM (Quanta 600, FEI). All images were acquired with high-resolution (2048 × 1768

pixels) at an accelerating voltage of 1 kV with a working distance of 5 mm using secondary electron detector to investigate surface nanoscopic features. No additional conductive coating was applied to any of the samples for the imaging.

## Electrochemical analysis

All electrochemical analysis was performed in 1× PBS, unless otherwise noted. Linear sweep voltammetry (LSV) measurement was carried out using a potentiostat (Gamry, Reference 600+) in three-electrode cell setup. To interface the electrode with electrolyte, a 3D-printed reservoir was glued on the catalytic array chip using polydimethylsiloxane (Dow Corning, Sylgard 184). Pt wire and Ag/AgCl electrodes (DRIREF-5SH, WPI Inc) served as counter and reference electrodes, respectively. LSV curves were collected from 0.7 V to 1.3 V versus 1 M Ag/AgCl at a scan rate of 10 mV s$^{-1}$. Based on the calibration of reference electrodes versus standard hydrogen electrode (SHE), the recorded potential was converted versus reversible hydrogen electrode (RHE) (Eq. 1), and overpotential was calculated using Eq. 2. Additionally, LSV measurements were also conducted in a two-electrode setup, employing Pt wire as a counter/reference electrode. Onset potential and compliance were determined from the x-intercept of the second derivatives of each LSV curve.

$$E_{vs.RHE} = E_{vs.SHE} + 0.0591 \times pH \qquad (1)$$

$$\eta = E_{vs.RHE} - 1.23 \qquad (2)$$

## Oxygen production measurements

An optical oxygen sensor (Presens, PM-PSt7) was used to measure produced oxygen in vitro. The sensor needle was replaced with a modified blunt needle, so the sensor was flush with the needle tip. The sensor tip was placed ~500 μm above the ecO$_2$ working electrode using a micromanipulator. The sensor was three-point calibrated by saturating in N$_2$, O$_2$, and air prior to all measurements. Oxygen sensing experiments were done inside a chamber with a controlled environment under continuous N$_2$ purge. Prior to the oxygenation, the electrolyte (~500 μL) was purged for at least 45 minutes using N$_2$ to reach near zero ppm oxygen. Oxygenators were then operated in either chronoamperometry or chronopotentiometry modes for oxygen production. For duty cycle dependance study, a voltage pulse train with varying duty cycle was applied.

## Oxygenation byproducts measurements

For hydrogen peroxide sensing, an ecO$_2$ electrode was used in two-electrode configuration in 0.5 mL 1x PBS solution. Prior to the measurement, the electrolyte was purged with N$_2$ for ~1 h. The electrode was then subject to oxygen production in chronoamperometry mode for ~24 h. The H$_2$O$_2$ concentration was measured using a hydrogen peroxide assay kit (Abcam, AB102500) according to the manufacturer's protocol. Briefly, 48 μL of the assay buffer and standard solutions were added to a 96-well plate, followed by 1 μL each of Oxired Probe and the developer solution. The mixture was agitated for about 5 minutes to ensure thorough mixing. It was then incubated for 1 hour at room temperature before the absorbance was read at 570 nm using a microplate reader (BioTek, Cytation 3). The manufacturer's standard protocol was used to calibrate the measurements.

For chlorine sensing, commercially available Cl strips (Supelco, 117925) were used. The steps followed are the same as the peroxide sensing. A SIROF electrode with 4 mm$^2$ was used as the working electrode, and a platinum wire was used as a counter electrode. After the chronoamperometry measurements, a Cl strip was immersed in the electrolyte for at least 20 s. The color change was then recorded using a camera under consistent lighting conditions. The image was then

processed using a custom Matlab program to compare against the standard color values provided by the manufacturer (Supplementary Code 1). The grayscale value was used to extrapolate the $Cl_2$ concentration. At least four different electrodes were tested for these measurements. All oxygenator measurements were carried out using Keithley 2614 SMU controlled using an open-source software SweepMe. The measured oxygenation data was later corrected using the three-point calibration data obtained earlier.

### ARPE-19 cell engineering

The parental cell line, ARPE-19 (CRL-230) was purchased from ATCC. To engineer cells expressing leptin APRE-19, cells were transfected with a CAG-Leptin plasmid using a piggybac transposase helper plasmid. ARPE-19 cells were seeded at a density of 300k cells/well of a 6 well plate and incubated overnight at 37 °C in a 5% $CO_2$ humidified atmosphere. Cells were then transfected using Lipoectamine 3000 according to the manufacturer's protocol (Catalog no. L3000015 Thermo Fisher Scientific) with the leptin and helper plasmid in a ratio of 2:1. Beginning 24 h after transfection the cells were selected for transgene expression using media with 2 µg mL$^{-1}$ of puromycin (Catalog no. A1113803 Thermo Fisher Scientific).

### ARPE-19 alginate cell capsule fabrication

To prepare for alginate encapsulation alginate was dissolved at 1.4% (w/v) in 0.8% saline and sterile-filtered through a 0.2-µm syringe filter and all buffers were prepared and sterilized by autoclaving and sterile filtering through a 0.2-µm vacuum filter. Cells were trypsinized and collected before being washed three times with calcium-free Krebs solution (4.7 mM KCl (P3911, Sigma Aldrich), 25 mM HEPES (H4034, Sigma Aldrich), 1.2 mM $KH_2PO_4$ (P5655, Sigma Aldrich), 1.2 mM $MgSO_4 \cdot 7H_2O$ (M2773, Sigma Aldrich), 135 mM NaCl (S9888, Sigma Aldrich)) by centrifuging the cells at $250 \times g$ for 5 min. After the third wash, the cells were resuspended in alginate at a concentration of 60k cells mm$^{-3}$. Encapsulation was done via electrospraying custom-built electrostatic spraying device. The device consisted of a syringe pumps (Harvard Apparatus), a 30 gauge blunt tip needle and a voltage generator (Gamma High Voltage) that was attached to the tip of the needle and grounded to a glass dish containing a cross-linking bath (20 mM barium chloride (342920, Sigma Aldrich), 5% mannitol (M9546, Sigma Aldrich)). Cell laden alginate droplets were expelled from the needle into the cross-linking solution at a rate of 3 ml h$^{-1}$. The size of the capsules was maintained by adjusting the voltage on the generator with a voltage of ~6.14 kV applied to produce capsules that were 300 µm in diameter. Capsules were incubated in the cross-linking bath for 10 min to ensure complete cross-linking. They were subsequently washed three times with Hepes buffer (0.132 M NaCl, 4.7 mM KCl, 1.2 mM $MgCl_2 \cdot 6H_2O$ (M2393, Sigma Aldrich), 25 mM HEPES) three times and transferred to a flask containing complete cell culture medium (Dulbecco's modified Eagle's medium (DMEM/F-12), with 10% fetal bovine serum (FBS) and 1% antibiotic-antimycotic (AA)) and maintained with standard cell culture techniques prior to experimental setup.

### On-site cell capsule oxygenation in vitro

The chip was wire bonded (West Bond, Wedge, Aluminum wire) to an inhouse designed printed circuit board (PCB) with a 16-pin connector, allowing for individual control of the devices. To hold media for cell cultivation, PDMS-passivated (Dow chemical, Sylgard 184) 3D-printed media reservoir was applied on catalytic arrays (Supplementary Fig. 8) to compartmentalize 8 devices per chip. Prior to the in vitro experiments, all components that were placed in an incubator were autoclaved (125 °C for 30 min). A volume of 250 µL of the basal medium DMEM:F-12 (Gibco, 11320033) formulated with 10% fetal bovine serum and 1% pen/strep was employed as a cell culture media. The assembled setup was used to extend the connection outside the incubator and

connect it to potentiostat (PalmSens 4, PalmSens) via 8-channel multiplexor (MUX8-R2, PalmSens). The devices were operated in chronoamperometry mode and voltage was gradually ramped to maintain a constant current. While an in vitro experiment was executed, media was exchanged every 3 days.

### Cell viability assay

Cell viability was determined using calcein acetoxymethyl (Calcein AM) and ethidium homodimer (InVitrogen, L3224) for live and dead cells, respectively. Nuclei were labeled using Hoechst 33342 (Thermo Fisher Scientific, 62249). Calcein AM, ethidium homodimer and Hoechst 33342 dyes were added with a final concentration of 2 µM, 4 µM and 1 µM, respectively, to each sample and incubated in a normal incubator for 30 min at 37 °C under 5% $CO_2$. The cells were then washed three times with warm 1× DPBS. Differential interference contrast (DIC) and scanning laser confocal fluorescence imaging were carried out using an upright confocal microscope and a ×20/0.50 NA water immersion objective (Nikon). Viability was quantified at least 4 samples per condition and 5 images per sample. Collected images were analyzed by Matlab code to count nuclei, live and dead cells (Supplementary Code 2). Note that Day 0 live/dead assay was performed prior to all long-term cultivation of capsules to exclude the influence on the viability other than incubation conditions. The viability was calculated as

$$Viability\,(\%) = \frac{(The\ total\ number\ of\ cells - The\ number\ of\ dead\ cells)}{(The\ total\ number\ of\ cells)} \times 100$$

(3)

To analyze the cell viability at single-capsule-level, we applied mask to define the footprint of each capsule onto the collected z-stack images. The mask was created using maximum-intensity projection (MIP) images of each z-stack image. Capsules in MIP images were defined as an individual capsule using image binarization and water shedding, and then labeled. Nuclei, live and dead cells were only counted within the defined capsule to analyze the viability at single-capsule-level along the z-axis. For additional information see Supplementary Figs. 9 and 10 and Supplementary Code 3.

### Leptin cell secretion assay

In order to evaluate peptide production, leptin ELISA kits (Abcam, Human leptin ELISA) were used. Following the manufacturer's guideline, 3 sets of standard leptin solutions (1000, 500, 250, 125, 62.5, 31.25, 15.63 and 0 pg mL$^{-1}$) were used to generate calibration curves in each round of analysis. The leptin-containing samples were collected 24 h after complete media exchange to evaluate 24-h peptide secreting capability. All samples were collected from 4 independent cultivation and technically triplicated with dilution factors of 100×, 500×, 1000×. The concentration of leptin was measured by recording the absorbance at 450 nm using a plate reader (Molecular Devices, SpectraMax i3X)

### Remote potentiostat

A compact (12 mm × 20 mm) printed circuit board was developed to provide the necessary voltage to the oxygenation electrode and to enable real-time in vivo monitoring of oxygen generation. The circuit features a Bluetooth Low Energy (BLE) enabled microcontroller, chosen for its low power consumption, and a dual-channel 12-bit buffered digital-to-analog converter (DAC) that allows for precise biasing of the catalysts. Four oxygen-generating channels and two calibration channels are connected in series to the DAC outputs through digitally controlled single-pole double-throw (SPDT) analog switches. The circuit is powered by a lithium polymer battery and supporting circuitry. One DAC channel continuously supplies the catalysts with a known voltage (the "oxygenation output"), while the other is wired in a force-

sense configuration with a feedback resistor (the "measurement output"). The microcontroller selects the DAC output for each of the four oxygen-generating channels. During normal operation, all channels are connected to the oxygenation output. Periodically, the microcontroller switches each channel in sequence to the measurement output, programmed to produce the same voltage as the oxygenation output. After a delay, the microcontroller's analog-to-digital converter (ADC) reads the voltage across the feedback resistor and calculates the current flowing into each channel. The calibration channels connect to precision (0.5%) resistors, facilitating measurement calibration throughout the experiment as conditions change. The voltage, measurement frequency, and delay can be wirelessly programmed in real-time using a companion mobile application. The microcontroller transfers measurement data to the application once its buffer is full, allowing the operator to view and adjust parameters for continued operation.

### in vivo experiments in rat model

A 2×4 array was fabricated and soldered to a PDMS-passivated gold-wire bundle which was then terminated with a 16-pin connector (Omnetics 2207). A PDMS housing with a custom mold was used to enclose the array, with each well holding a 2×4 array. The assembly was then soaked in 70% ethanol for 24 h, followed by deionized water for another 24 h, with water exchanges every 8 h. A commercially available polycarbonate membrane (Sterlitech PCTF 0425100) with 0.4 μm pore size was attached to the top of the well to provide immunoisolation while facilitating nutrient and oxygen exchange. $ecO_2$ devices for in vivo oxygenation were sterilized using ethylene oxide (ETO) sterilization (Anprolene AN75, Anderssen) and allowed to rest for 48 h. Maintaining aseptic conditions, ARPE-19 cell capsules were transferred into the cell compartment using a pair of needles. Note that cell capsules were resuspended in fresh media and approximately 800 capsules were loaded. Note that media exchange was unable due to inaccessibility to the device during implantation. Prior to surgery, a SAS Sprague Dawley rat was anesthetized using isoflurane in the chamber and it was maintained via nose cone delivery. Note that sex was not considered in the study design or analysis as it did not involve any disease models. Shaved rat skin was sanitized using IPA. To minimize mechanical stress on skin and bones, $ecO_2$ device was implanted in the abdominal skin pocket. The remote potentiostat was enclosed in 3D-printed housing and placed on the backside to prevent damage from rat's motion. Those electronics were connected using flexible 16-channel gold wire, which was routed under the skin. Electrochemistry of $ecO_2$ was wirelessly controlled using an application (Flexi-BLE) on an iOS device. Cell viability was performed following the above-mentioned procedures (Cell viability assay).

### Statistics and reproducibility

For statistical analysis, we predetermined effective sample size based on power analysis, executing all in vitro and in vivo experiments with biologically independent replications of >4. In case of data exclusion, we determined the outlying data points based on box plot analysis. Data points lied on >1.5 interquartile range were excluded. The experiments were not randomized, and the investigators were not blinded to allocation during experiments and outcome assessment.

### Reporting summary

Further information on research design is available in the Nature Portfolio Reporting Summary linked to this article.

## Data availability

All data supporting the findings of this study are available within the article and its Supplementary Files. Any additional requests for information can be directed to, and will be fulfilled by, the corresponding authors. Source data are provided with this paper.

## Code availability

The codes that the authors used for data analysis are available in Supplementary Code or from the corresponding authors on reasonable request.

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

## Acknowledgements

This material is based on research sponsored by 711 Human Performance Wing (HPW) and Defense Advanced Research Projects Agency (DARPA) under agreement number FA8650-21-1-7119. The U.S. Government is authorized to reproduce and distribute reprints for Governmental purposes notwithstanding any copyright notation thereon. The views and conclusions contained herein are those of the authors and should not be interpreted as necessarily representing the official policies or endorsements, either expressed or implied, of 711 Human Performance Wing

(HPW) and Defense Advanced Research Projects Agency (DARPA) or the U.S. Government. T.C.-K., S. Jo, and I.L. acknowledge support from Carnegie Mellon University Department of Materials Science and Engineering Materials Characterization Facility supported by Grant MCF-677785. T.C.-K., S. Jo, and I.L. would like to thank Dr. Loren Rieth for helpful discussion about SIROF and Mark Weiler and Dr. Matthew Moneck from the Claire and John Bertucci Nanotechnology Laboratory for assistance with SIROF deposition. J.R., A. Surendran, and X.J. acknowledge Northwestern University's NUANCE Center, which has received support from the SHyNE Resource (NSF ECCS-2025633), the IIN, and Northwestern's MRSEC program (NSF DMR-1720139). J.R. and A. Surendran acknowledge support from Analytical BioNanoTechnology Equipment Core (ANTEC) supported by the Soft and Hybrid Nanotechnology Experimental (SHyNE) Resource (NSF ECCS-2025633) for material characterization and Center for Advanced Microscopy/Nikon Imaging Center (CAM) supported by CCSG P30 CA060553 awarded to the Robert H Lurie Comprehensive Cancer Center for confocal imaging. J.R., T.C.-K., A. Surendran, and I.L. acknowledge support from Blackrock Neurotech for SIROF, and wire-bundle bonding for the in vivo experiments. D.J.S. was supported by the National Heart, Lung, and Blood Institute of the National Institutes of Health under Award Number K99HL155777. The content is solely the responsibility of the authors and does not necessarily represent the official views of the National Institutes of Health.

## Author contributions

T.C.-K. and J.R. conceived the work. I.L., A. Surendran, S. Jo, G.N., A. Sipe, and X.J. performed materials synthesis, characterization, and down selection. I.G., T.S., and I.L. performed finite element modeling of ecO$_2$. I.L., A. Surendran., and G.N. and S. Jo fabricated and tested electrochemical performance, oxygenation, and side product formation of ecO$_2$ devices. I.L., A. Surendran., and A.S.M. performed in vitro cell studies and viability studies. S.F., C.F., and O.V. performed cell engineering work and cell encapsulation. A.C., B.R., and J.H. designed and deployed the wireless potentiostat for in vivo experiments. D.J.W., O.R., S. John, and I.L. performed in vivo implantation in rodents. D.J.S., Y.W., and A.W.F. assisted with analysis of cell imaging data and visualization. J.H., D.J.W., O.V., A.F., J.R., and T.C.-K. oversaw the research. I.L., A. Surendran, J.R., and T.C.-K. wrote the manuscript. All authors revised and approved the manuscript.

## Competing interests

I.L, A. Surendran, S. Jo, X.J., T.C.-K., and J.R. are co-inventors on a U.S. provisional patent application with application serial number US2023/ 019215 jointly filed by Carnegie Mellon University, and Northwestern University. J.R., T.C.-K. and O.V. are co-founders of a company, FlashBio Inc. The other authors declare no competing interests. The invention covers the aspects of localized, multisite and readily controlled oxygen supply for cell cultures. The device aspects described herein are included in the patent.
