## [Peer review file · Nature Communications]

REVIEWER COMMENTS

Reviewer #1 (Remarks to the Author):

The authors describe the development of a novel chip to enable delivery of oxygen to implanted cells. The device is tested extensively in biological contexts in the manuscript, including characterization of off-target products in such a system. Though this work is interesting and would be important to the diverse readership of the journal, we would like to see additional chemical characterization of oxygen diffusion following production in the electrochemical chip. We would therefore suggest accepting the manuscript following the additional proposed experiment.

Though the authors only report side product production electrochemically based on applied potential and not based on distance from the electrode. We would additionally appreciate off-target screens for components of the mixture that may be generated electrochemically, namely additional ROS species. Beyond additional biochemical characterization, we would suggest removing some of the "jargon" from the introduction and checking to ensure consistency between figure captions and what they claim to show.

Reviewer #2 (Remarks to the Author):

The idea of immuno-isolated cell therapy has been around for several decades (with primary application in diabetes). However, for a variety of reasons which the authors alluded to, this approach has not been successful and has not made any clinical inroads. One of the reasons has been the difficulty of keeping the cells alive and well oxygenated. Although the authors discuss a method for in situ oxygen generation through water electrolysis, the reviewer is skeptical that this method will become a clinical reality and will have a practical impact. Recent rapid adoption of CGMS and pump therapy has made the encapsulated cell therapy solution to management of diabetes very unlikely. The required power source (in this case battery) and energy requirements to keep cells alive for an extended period of time makes this approach impractical. However, transient increase in oxygen tension in hypoxic tumors as was illustrated by Maelki et al (Maleki, Teimour, et al., "An ultrasonically powered implantable micro-oxygen generator (IMOG)." IEEE transactions on Biomedical Engineering 58, no. 11 (2011): 3104-3111.) can benefit from such approach.

Reviewer #3 (Remarks to the Author):

The manuscript titled 'Electrocatalytic on-site oxygenation for transplanted cell-based-therapies' has successfully developed the exogenous oxygen production devices which can provide oxygen using an electrocatalytic approach. A highly active and biocompatible electrocatalyst, nanostructured iridium oxide (SIROF), has been employed into the oxygenation device system for electrochemical water electrolysis, creating an effective, stable, and safe on-site oxygen delivery tool. The manuscript is well written and interesting. The device can clearly overcome the delay or lack of vascularization from the transplant host to newly transplanted cells. The manuscript is very timely and provides an interesting approach. Some comments that should be addressed prior to publication are:

1. The authors mention in the abstract 'Previous oxygenation strategies have targeted gas circulation or decomposition of solid peroxides. These strategies however require bulky implants, transcutaneous supply lines, and are limited in their total oxygen production or regulation'. How does the size of the mentioned implants compare to the size of the device reported in the manuscript?

2. How was the cell density chosen? Is this cell density representative of a therapeutic treatment? Which one?

3. What was the limit of cells vs oxygen provided tested? I believe this would be interesting for the reader to know the maximum capabilities of the device.

4. The FEA simulation shows the oxygen distribution considering free space however, was this simulation also performed considering the cell spheres? What proportion of the space is occupied by these cellular spheres in relation to the total available volume on the device? How does alginate interplay with the oxygen diffusion? Could the authors perform the FEA or comment on this on the manuscript?

5. Fig 1 shows the cell spheres densely packed inside the device, how does the viability of cell in spheres at the top section of the device compare in relation to the spheres that received oxygen (according to the distance in Fig1b)? Please add clarification regarding this and which electrode the y axis is referring to.

6. Could the authors comment on why the in vitro analysis was performed over 21 days but the in vivo was only performed over 10 days? Is 10 days oxygenation enough to meet the cellular metabolic demands in vivo? Please provide evidence of this particularly considering the authors' statement that vascularization requires 14 days. Please clarify.

7. Nanostructure of SIROF was examined using SEM before or post duty cycle. The images of SEM in high magnification were blurry. Please add the methods of SEM and duty cycles.

8. For the LSV scan, the 1.6 V of potential voltage was used in 3-electrode setup while 2-electrode setups used 1.7V. Can the authors add the clarification for this? Also, can the authors indicate the applied voltage used for the subsequent in vivo and in vitro experiments?

9. For how long was the media de-oxygenated? Was there any FEA based analysis to approximate the de-oxygenation or was it based on measurements? Low oxygen levels should be below 2%. Please clarify the reason of why you chose 1% oxygen instead of 2% or completely removing the oxygen (0% oxygen) in the manuscript and elaborate how to prepare the deoxygenated media.

10. Authors have cultured cells at normal oxygen level and found that cell death, indicating hypoxic death. Is it possible that the cell death occurred due to the pre-existing cell death or the effect of the alginate on cells? Were the cells characterized or cell viability assayed before mixing with the alginate? Please add those results as well as discuss the effect of alginate on cell viability?

11. The authors indicated that there was no significant inflammation and foreign body response post in vivo study. Is there any data of immune cells in support of this statement?

12. The authors reported that the varied cell viability between in vitro and in vivo was due to the different volume of media per cell. Please add the media for in vivo study in methods and clarify whether the media were used and changed in the in vivo study.

Minor comments:

Typo on Supplementary Fig 24 & 25 'what are the colors' at the caption has been left from the corrections.

Dense cells were encapsulated into chips as PDMS enclosure applied. There was no reaction or integration with host tissue, which limits the implications of in vivo applications. Please address and clarify this assumption for the in vivo model.

The safety and accessibility of this system plays the important role in transplanted cell-based-therapies. A battery-free wireless power can make the system more flexible rather than battery power.

and connecting cables. Please comment on this.

For the in vivo, the channels have not been compartmentalized. Is it possible to enlarge size of arrays for producing large amounts of oxygen? Please comment on this in future direction.

Reviewer #4 (Remarks to the Author):

I co-reviewed this manuscript with one of the reviewers who provided the listed reports as part of the Nature Communications initiative to facilitate training in peer review and appropriate recognition for co-reviewers.

Response to reviewers' feedback.

In the below response letter, the comments/questions of the editors or reviewers are in black. Responses are in blue. Changes to the main text or SI are highlighted in yellow.

Reviewer #1 (Remarks to the Author):

The authors describe the development of a novel chip to enable delivery of oxygen to implanted cells. The device is tested extensively in biological contexts in the manuscript, including characterization of off-target products in such a system. Though this work is interesting and would be important to the diverse readership of the journal, we would like to see additional chemical characterization of oxygen diffusion following production in the electrochemical chip. We would therefore suggest accepting the manuscript following the additional proposed experiment.

- 1. Though the authors only report side product production electrochemically based on applied potential and not based on distance from the electrode. We would additionally appreciate off-target screens for components of the mixture that may be generated electrochemically, namely additional ROS species.*

We thank the reviewer for the comment about the generation of other reactive oxygen species (ROS) in our electrocatalytic reaction platform. We have taken your suggestion seriously and conducted a deep literature study of other ROS that could be generated in our system. We have now added additional references to the manuscript to represent the additional ROS reported in similar electrocatalytic devices. The reviewer's concern of other ROS generated in our study is addressed in the following response. The additional ROS we identified are unlikely to be generated in our system at significant levels. Additionally, most of these ROS are unstable at neutral pH, and undergo disproportionation and dismutation reactions and end up as hydrogen peroxide in the electrolyte. We believe that the benefits of our system outweigh the risks, but we want to be transparent to the readers about the potential risks.

- 1. Ozone production (O₃):** Anode materials with high oxygen overpotential (Pt, PbO₂, glassy carbon etc) are preferable to produce ozone through electrolysis of water¹. This is because they inhibit oxygen evolution reaction, which is more favorable than ozone evolution. Increased activation energy for OER reduces the reaction rate and allows more water molecules to be oxidized to form ozone as shown in the equation below.

Besides, the catholytes and anolytes need to be separated using a membrane to prevent mixing and avoid side reactions that reduce the efficiency of ozone generation. In the reported literature, ozone production is reported at voltages higher than 10 V.

Our ecO₂ platform uses Iridium oxide as the electrocatalyst, with a low over potential of 395±18 mV compared to platinum electrode's 565±10 mV. Iridium oxide favors oxygen evolution (2H₂O→O₂+4H⁺+4e⁻), as the activation energy is lower. Probability of ozone production is further impeded as the cathode and anode are not separated in the ecO₂ platform. Finally, ecO₂ is operated at a safe operation window of 1.6 V to 1.9 V that favors oxygen evolution reaction (OER) over ozone evolution.

- 2. Superoxide ion (O₂⁻):** Formation of superoxide ion in water electrolysis depends on the electrolyte pH, and activity and stability of anode materials. While anode materials with high

activity and low over potential such as IrOx can promote the formation of O_2^- , it can easily undergo disproportionation and dismutation reactions to form other products such as hydrogen peroxide (which we measure) and oxygen^{2,3}. An alkaline pH of ≥ 11 is more favorable for the stability of superoxide ions. Therefore, most of the superoxide ions will be consumed and will not accumulate in the neutral electrolyte we report in this study. In addition, most biological cells contain superoxide dismutase enzymes that catalyze the dismutation of superoxide anions.

- 3. Hydroperoxide ion (HO_2^-):** Formation of hydroperoxide is also favored by materials with high over-potential that inhibit four-electron reduction of oxygen to water. In a buffered solution at neutral pH such as 1xPBS/human blood, hydroperoxide will be protonated to form H_2O_2 (which we measure)^{4,5}. It may rarely form unstable compounds with phosphates in PBS solution such as peroxophosphate ($H_2PO_5^-$) and peroxyphosphate ($H_2PO_4^-$). However, if formed, they are highly unstable at neutral pH, and decompose to phosphate and hydrogen peroxide.
- 4. Singlet oxygen (O_2^*):** Anode materials with low overpotential such as Iridium oxide could generate singlet oxygen, which is an excited state of molecular oxygen. But in neutral pH it is highly unstable and participates in hydrolysis reaction and forms the aforementioned byproducts such as superoxide ion, hydroperoxide ion, peroxophosphate, and hydrogen peroxide^{6,7}. As discussed above, these byproducts further undergo decomposition and form mostly hydrogen peroxide and oxygen.
- 5. Hydroxyl radical (OH^*):** One-electron oxidation of water molecules may produce highly reactive and short-lived hydroxyl species at anode. They rapidly undergo recombination, disproportionation, hydration, or oxidation reactions in neutral pH and form aforementioned byproducts, with the most common byproduct being hydrogen peroxide^{8,9}.
- 6. Hydrogen peroxide (H_2O_2):** Hydrogen peroxide production in electrochemical hydrolysis of water follows a two-electron oxygen reduction at the cathode. Equilibrium potential for the hydrogen peroxide production shifts to more positive values with increasing pH values (basic), even though the overpotential for this reaction is also increased, resulting in slower kinetics¹⁰. Our work demonstrates electrochemical oxygen production at neutral pH, which is less favorable for 2e-ORR at cathode as opposed to the 4e pathway of ORR. Moreover, a moderate bond strength¹¹ between $*OOH$ intermediate and the electrocatalyst is favorable for H_2O_2 production as it enables a balanced adsorption and desorption of $*OOH$ intermediate at the catalyst surface. The strong bond strength of iridium oxide with this intermediate favors the formation of oxygen and water, as it lowers the activation energy for O-O bond cleavage. In addition, the electrode system in ecO₂ could enable the decomposition of H_2O_2 at anode ($2H_2O_2 \rightarrow H_2O + O_2$) and reduction at cathode ($H_2O_2 + 2H^+ + 2e^- \rightarrow 2H_2O$), further limiting H_2O_2 production¹². As evidenced in our study, hydrogen peroxide production is more evident beyond 1.9 V. We therefore limited the operation window of ecO₂ to 1.6 V to 1.9 V.

In conclusion, while other reactive oxygen species (ROS) are possible, they are either highly unfavorable to form at neutral pH or are likely to decompose to hydrogen peroxide (H_2O_2). While we acknowledge the possibility of these side reactions, we stress that these ROS are not a major concern in the context of our study. This is further supported by our 21-day *in vitro* study, which evidenced only 7.5% cell death using our

ecO₂ platform in hypoxia (1% oxygen), in contrast to 91.4% cell death without ecO₂. This suggests that hypoxia is the primary cause of cell death, and that any ROS generated by the ecO₂ platform do not have a significant impact. The references are given at the end of this document.

Changes to the main text are appended.

Added to the main text: "While there is a possibility of generating other reactive oxygen species (ROS) such as hydroperoxide ion, superoxide ion, hydroxyl radical, and singlet radical, these ROS are not a major concern in the context of the current study. These are unstable at neutral pH and undergo disproportionation or dismutation reactions and end up as hydrogen peroxide in the electrolyte^{36–38}."

New references added to the main text:

36. Lyle, H., Singh, S., Paolino, M., Vinogradov, I. & Cuk, T. The electron-transfer intermediates of the oxygen evolution reaction (OER) as polarons by *in situ* spectroscopy. *Physical Chemistry Chemical Physics* **23**, 24984–25002 (2021).

37. Naito, T., Shinagawa, T., Nishimoto, T. & Takanabe, K. Recent advances in understanding oxygen evolution reaction mechanisms over iridium oxide. *Inorganic Chemistry Frontiers* vol. 8 2900–2917 Preprint at <https://doi.org/10.1039/d0qi01465f> (2021).

38. Chen, G. *et al.* A discussion on the possible involvement of singlet oxygen in oxygen electrocatalysis. *Journal of Physics: Energy* **3**, 031004 (2021)."

2. *Beyond additional biochemical characterization, we would suggest removing some of the "jargon" from the introduction and checking to ensure consistency between figure captions and what they claim to show.*

We thank the reviewer for the feedback. We made the following changes to the introduction.

Removed allogeneic or xenogeneic.

"The transplantation of therapeutic cells, within semipermeable devices as a living pharmacy has the potential to treat a range of diseases such as endocrine disorders, autoimmune syndromes, cancers, and neurological degeneration".

Replaced neovascularization with "the formation of new blood vessels".

Replaced size-exclusive membranes with "size-selective membranes".

Corrected the **Figure 3b** caption in the main text. Replaced immunofluorescence with "Live/dead assay fluorescence images".

Corrected the **Figure 4c** caption in the main text. Replaced immunofluorescence with "Live/dead assay fluorescence images".

Reviewer #2 (Remarks to the Author):

The idea of immuno-isolated cell therapy has been around for several decades (with primary application in diabetes). However, for a variety of reasons which the authors alluded to, this approach has not been successful and has not made any clinical inroads. One of the reasons has been the difficulty of keeping the cells alive and well oxygenated. Although the authors discuss a method for in situ oxygen generation through water electrolysis, the reviewer is skeptical that this method will become a clinical reality and will have a practical impact. Recent rapid adoption of CGMS and pump therapy has made the encapsulated cell therapy solution to management of diabetes very unlikely. The required power source (in this case battery) and energy requirements to keep cells alive for an extended period makes this approach impractical. However, transient increase in oxygen tension in hypoxic tumors as was illustrated by Maelki et al (Maleki, Teimour, et al., "An ultrasonically powered implantable micro-oxygen generator (IMOG)." IEEE transactions on Biomedical Engineering 58, no. 11 (2011): 3104-3111.) can benefit from such approach.

We appreciate your time and effort in reviewing our manuscript. We have addressed your concerns in the revised manuscript, and we are confident that it has improved the manuscript significantly.

1. "Although the authors discuss a method for in situ oxygen generation through water electrolysis, the reviewer is skeptical that this method will become a clinical reality and will have a practical impact. Recent rapid adoption of CGMS and pump therapy has made the encapsulated cell therapy solution to management of diabetes very unlikely. "

Our work presents a compelling proof-of-concept, showcasing the ability of a new electrocatalytic platform to maintain cell viability and functionality without any harmful side effects. We note that while diabetes treatment is one possible application area, there are many others as noted. We would like to clarify that the high cell density in the presented work was mainly used to benchmark the performance of our platform for future applications. While the platform can be used to support high-density, biologically engineered, therapeutic cells for applications such as regenerative medicine and treatment for intractable diseases¹³, future applications of this technology are broad and impactful.

2. "The required power source (in this case battery) and energy requirements to keep cells alive for an extended period of time makes this approach impractical. "

We would like to emphasize that the battery was used in the current iteration of ecO₂ to primarily demonstrate the technology *in vivo*, and for simplicity. We anticipate our approach is not limited to the current setup and can benefit from various existing and emerging technologies in wireless power transfer and communication, e.g., magnetoelectrics¹⁴, or on-site energy harvesting¹⁵.

Regarding the concern about the power requirements of our system. We would like to clarify that the average current required by the supporting circuitry to operate is only ~34.93 μ A at 2.2 V. In our *in vitro* and *in vivo* studies, we varied the potential of the oxygenator chips to maintain an average current of about 500 nA. This means that the lithium-ion batteries we used in this study, with a peak voltage of 3.7 V and a current rating of 140 mAh, would have a battery life of about 165 days before needing recharge. Further, in a future iteration, we are working on adding additional power saving features for our supporting circuitry to extend the lifetime in a smaller form factor, including a completely battery free approach.

3. "However, transient increase in oxygen tension in hypoxic tumors as was illustrated by Maelki et al (Maleki, Teimour, et al., "An ultrasonically powered implantable micro-oxygen generator (IMOG)."

IEEE transactions on Biomedical Engineering 58, no. 11 (2011): 3104-3111.) can benefit from such approach.”

We agree that our ecO_2 platform could be used to address the low oxygen tension in tumors and prevent abnormal vascularization and inhibit the expression of HIF-1 with reduced needed power due to the use of electrocatalyst.

Reviewer #3 (Remarks to the Author):

The manuscript titled 'Electrocatalytic on-site oxygenation for transplanted cell-based-therapies' has successfully developed the exogenous oxygen production devices which can provide oxygen using an electrocatalytic approach. A highly active and biocompatible electrocatalyst, nanostructured iridium oxide (SIROF), has been employed into the oxygenation device system for electrochemical water electrolysis, creating an effective, stable, and safe on-site oxygen delivery tool. The manuscript is well written and interesting. The device can clearly overcome the delay or lack of vascularization from the transplant host to newly transplanted cells. The manuscript is very timely and provides an interesting approach. Some comments that should be addressed prior to publication are:

1. The authors mention in the abstract 'Previous oxygenation strategies have targeted gas circulation or decomposition of solid peroxides. These strategies however require bulky implants, transcutaneous supply lines, and are limited in their total oxygen production or regulation'. How does the size of the mentioned implants compare to the size of the device reported in the manuscript?

We thank the reviewer for their comment. In table 3.1, we compare the size of our implant with various other techniques reported in the literature for oxygenation *in vivo*. In this manuscript, we present a compelling proof-of-concept where the oxygenator itself is made in a thin film form factor that is 4mm² in area. The number of devices in ecO₂ can be varied depending on the specific requirements of the therapy, such as the desired cell volume or total cell number. Therefore, the size of the device could be varied significantly based on its intended application. The electronics, PCB, and battery may be replaced with more compact or battery-free alternatives. Further, from the perspective of implant size, the currently reported ecO₂ is comparable if not smaller than many clinical device implants, like intrathecal pumps (Synchromed II: 86 mm \varnothing ×19.5mm), and stimulators (Reactiv8: 48mm×65mm×12mm).

Table 3.1: A comparison of the sizes of various implantable oxygenation devices reported in the literature.

Strategy	Citation	Implant dimension	Remarks
Direct gas delivery (BetaO2)	16	68 mm ϕ ×18 mm thickness and 100 mm and external reservoir + access ports	Active delivery. Continuous gas supply required from an external oxygen reservoir through access ports. On-demand oxygen production.
Microfabricated electrochemical oxygen generator	17	10 mm×10 mm chips	Active delivery. Bubble formation increases risks of gas embolism. On-demand oxygen production.
Ultrasonically powered electrochemical oxygen generator	18	1.2 mm×1.3 mm×8 mm	Active delivery. Platinum CE and WE. High potential for oxygenation. On-demand oxygen production.
Superoxide scaffold	19	5 mm ϕ ×1.2 mm	Passive delivery. Not a chronic solution as the oxygen concentration reduces with time. Not on-demand oxygen production. Not on-demand oxygen production.
Inverse-breathing superoxide	20	1.96 mm ϕ ×2 cm	Passive delivery. Predetermined oxygen concentration. Not on-demand oxygen production.
ecO ₂	This work	19×9×3mm	Active delivery. In vivo ecO ₂ , controllable, on-demand oxygen production.

2. How was the cell density chosen? Is this cell density representative of a therapeutic treatment? Which one?

Previously published work demonstrated survival of implanted encapsulated cells at densities of 6-10k cells/mm³ for treating type 1 diabetes and psoriasis (see citations 15 and 16 in our manuscript ^{21,22}). To robustly test the efficacy of our technology, we chose a cell density 6-10 fold above previously published work. At 60k cells/mm³ density that we used; majority of the cells die within 21 days (without oxygenation) allowing us to demonstrate prevention of cell death. Additionally, clinically tested cell laden devices are

relatively large (68 mm diameter for BetaO2 devices and 9×3 cm for ViaCyte devices²¹. Increasing cell density using our ecO₂ platform would enable smaller devices to contain the same number of therapeutic cells.

We added this point in the introduction of our manuscript:

"Studies have shown that encapsulated cells can survive at 6-10k cells/mm³ when implanted for treating type 1 diabetes and psoriasis^{10,11}."

3. What was the limit of cells vs oxygen provided tested? I believe this would be interesting for the reader to know the maximum capabilities of the device.

We appreciate your feedback. As mentioned in the above response, we tested 60k cells/mm³, which is six times the maximum cell density reported in the literature for similar applications^{21,22}. The alginate encapsulation technique we used in our work is limited to a maximum cell density of 60k cells/mm³. While pushing the limits of ecO₂ is an important aspect of the technology development, the current work serves as a proof-of-concept for electrocatalytic oxygenation for cell therapeutics. In a future iteration of this work, we plan to achieve higher cell densities closer to tissues using cellular scaffolds. This will necessitate the development of different geometries for ecO₂, such as 3D oxygenators to allow more uniform oxygenation, negating the surface-level effects. Mammalian cells require oxygen and nutrients for survival and are located within 100 μm to 200 μm away from blood capillaries²³. This suggests that ecO₂ V2.0 could potentially facilitate tissue level oxygenation. However, these advanced ideas are beyond the scope of this work and will be pursued in future.

4. The FEA simulation show the oxygen distribution considering free space however, was this simulation also performed considering the cell spheres? What proportion of the space is occupied by these cellular spheres in relation to the total available volume on the device? How does alginate interplay with the oxygen diffusion? Could the authors perform the FEA or comment on this on the manuscript?

We thank the reviewer for this comment. The motivation of FEA simulation is to assist designing microfabricated oxygen evolution catalysts arrays producing uniform oxygen distribution (unlike a single site that will have a Gaussian distributed oxygen diffusion profile). Therefore, our focus on the simulation studies was the generated oxygen profiles from ecO₂ devices, not simulating electrochemical oxygen supply with cell capsules. Indeed, the tools we developed can be used to simulate oxygen profiles with non-uniform media, i.e. with alginate capsules on the ecO₂. Supported by literature for diffusional behavior studies in alginate spheres²⁴, we additionally executed diffusion simulation studies as shown in **Response Figure 4-1**. While the diffusion rate is relatively higher in the model without capsules (**Response Figure 4-1a and c**), the uniform distribution tendency, however, is still maintained in the model (**Response Figure 4-1b and d**) with a monolayer of capsules. Although oxygen is localized in the capsules to some extent because of reduced oxygen diffusivity in Ba²⁺-alginate that we employed in the studies, the variance of oxygen concentration was not greater than 15 % within capsules and in comparison, between Ba²⁺-alginate capsules and water. Additionally, the free space between capsules contributed to alleviated oxygen localization in the model having capsules with faster oxygen diffusivity. Given a relatively shorter period (300 and 3,000 sec) compared to the actual cell oxygenation (up to 21 days), it is clear that the compartment will be uniformly saturated as oxygen is continuously supplied, implied by the change of oxygen distribution between simulation for 300 sec and for 3,000 sec. In cell compartment of the implanted ecO₂ device, a monolayer of the capsules (ca. 800 capsules, with diameter of 400 μm) occupied ~5.23 % of the total volume.

We have modified the manuscript in text De Novo Design of ecO₂ section accordingly-

"To ensure uniform oxygen delivery to the implanted cells, we designed various array geometries and estimated the distribution of the generated oxygen from them, guided by finite element analysis (FEA)."

Response Figure 4-1. Oxygen diffusion simulation with or without Ba²⁺-alginate capsules at different time points. Diffusion simulation for 300 sec (a) without and (b) with Ba-alginate capsules with 40-µm gaps. Diffusion simulation for 3,000 sec (c) without and (d) with Ba-alginate capsules with 40-µm gaps; circles are Ba-alginate spheres. Oxygen concentration was normalized; scale bars are 200 µm.

5. Fig 1 shows the cells spheres densely packed inside the device, how does the viability of cell in spheres at the top section of the device compares in relation to the spheres that received oxygen (according to the distance in Fig1b)? Please add clarification regarding this and which electrode the y axis is referring too.

We thank the reviewer for the comment. For either *in vitro* or *in vivo* oxygenation, the ecO₂ was loaded with either 100 capsules or 800 capsules per well, respectively. Generally, the capsules formed a monolayer of densely packed capsules. At the end of the experiments, we retrieved the cell capsules from the ecO₂ device and transferred them to a 3D-printed compartment having transparent bottom that was used to image using scanning laser confocal microscopy followed by post imaging analysis. For this reason, the capsules were randomly aligned and stacked when we were performing post-oxygenation live/dead analysis so that we could not recognize where the top and bottom had been. The procedures of cell capsule viability assay are described in **Response Figure 5-1**. We note that the profile presented in **Figure 1b** was computed assuming cross-sectional oxygen distribution owing to the symmetry of the designed geometry. The position of the cross-section is described in **Supplementary Figure 1a**. To allow us to explore the effect of distance from the oxygenator on cellular viability, we performed live/dead analysis at single-capsule-level to have better insight of the efficacies of electrocatalytic oxygenation (**Response Figure 5-2**). Our empirical results verified

our system in terms of the efficacy of electrocatalytic oxygen delivery toward the encapsulated cells. As described in **Response Figure 5-3**, single-capsule level viability analysis showed that hypoxic core formation at the center of capsules was addressed with ecO_2 , while hypoxic incubation without oxygenation represented significantly compromised cell viability at the center in comparison to the viability at the edges. This result is visualized in **Response Figure 5-3a and b**, depicting greater cell viability at the surface of capsules in hypoxic incubation without ecO_2 . This observation further supports that the amount of produced oxygen is sufficient to address hypoxic cell death caused by the diffusion limitation, and the variance of oxygenation was negligible in 400- μ m capsules as we presented in the simulation results.

We have modified the manuscript in the Materials and Methods section accordingly-

"To analyze the cell viability at single-capsule-level, we applied masks to define the footprint of each capsule onto the collected z-stack images. The mask was created using maximum-intensity projection (MIP) images of each z-stack image. Capsules in MIP images were defined as an individual capsule using image binarization and water shedding, and then labeled. Nuclei, live and dead cells were only counted within the defined capsule to analyze the viability at single-capsule-level along the z-axis. For additional information see **Supplementary Figure 9 and 10.**"

We have added **Response Figure 5-1, 5-2 and 5-3** into the Supplementary Information accordingly-

"Response Figure 5-1 has been added in the Supplementary Information as **Supplementary Figure 9.**"

"Response Figure 5-2 has been added in the Supplementary Information as **Supplementary Figure 10.**"

"Response Figure 5-3 has been added in the Supplementary Information as **Supplementary Figure 16.**"

Response Figure 5-1. A schematic illustration of live/dead assay after *in vitro* and *in vivo* oxygenation with ecO_2 . (a) Device assembly – media reservoir was attached on microfabricated ecO_2 chip. Note that PDMS encapsulation and immune protection membrane were additionally applied for implantable devices. (b) Cell transfer – alginate cell capsules containing ARPE-19 cells were transferred into the media reservoir with the media. (c) Cell retrieval – after oxygenation, the capsules were collected and hoechst/calcein/ethidium stained to visualize nuclei, live and dead cells. For details, see Materials and Methods. (d) Cell imaging – the stained cells were z-stack imaged using fluorescence confocal microscopy. (e) Image analysis – the recorded z-stack images were analyzed in terms of viability.

Response Figure 5-2. Mask creating for single-capsule-level viability assay. (a) MIP image creation – z-stack images were projected in the xy-plane using maximum intensity projection (MIP). (b) Defining capsule footprint – the footprint of each capsule was defined using a watershed of binarized MIP images. (c) Labeling capsules – the defined capsules were labeled; scale bars are 200 μm .

Response Figure 5-3. Single-capsule-level viability assay of 21-day *in vitro* oxygenation. (a) Half-violin plots for the viability of individual capsules; red – hypoxia with ecO_2 ($n=158$); blue – hypoxia without ecO_2 ($n=122$); box – standard deviation; bar – mean; whisker with caps – 1.5 IQR (Interquartile range) (b) statistical analysis of viability at edges (top: 240 μm , bottom: -240 μm) and the center of capsules (center: 0 μm); ****: $p < 0.0001$; results are presented with mean \pm SD; box – standard deviation; bar – mean; whisker with caps – 1.5 IQR.

We have modified the **Figure 3** in the main text accordingly-

Figure 3. ecO₂ supports cell capsules *in vitro* for 21 days. (a) A schematic illustration of the *in vitro* oxygenation assay and analysis. (b) A representative set of 3D-rendered Live/dead assay fluorescence images after 21-day *in vitro* oxygenation; i - a representative 3D-rendered z-stacked images, ii - Expanded view of the marked white dashed circle, live cells, iii - dead cells. All scale bars are 100 μm (c) Cellular viability as a function of time, (d) Single-capsule-level spatial distribution of cell viability in cell laden alginate capsules. (e) A representative live/dead assay at single-capsule-level of 21-day hypoxic incubation with or without oxygenation; i - z = -240 μm; ii - z = 0 μm; iii - z = 240 μm; scale bars are 100 μm. (f) Peptide production as a function of *in vitro* culturing time. Blue corresponds to 1 % oxygen with ecO₂ (oxygenation in hypoxia, experiment, n=4); red corresponds to 1 % oxygen without ecO₂ (hypoxic incubator, negative control, n=4); ***: p < 0.001; ****: p < 0.0001.

We have modified the manuscript in text the ecO₂ sustains high cell density *in vitro* section accordingly-

"Cell viability distribution along the z-axis showed no hypoxic core formation for the cell-loaded capsules that were oxygenated with ecO₂, representing negligible variation between the edges and core. This insignificant difference implied that the cell death in these samples was not due to limitation in oxygen diffusion. In contrast, the samples without ecO₂ showed a clear cell death predominantly away from the surface, evidencing that diffusion limitation caused cell death with the formation of hypoxic core (Figure 3d, e and Supplementary Figure 16)."

6. Could the authors comment on why the *in vitro* analysis was performed over 21 days but the *in vivo* was only performed over 10 days? Is 10 days oxygenation enough to meet the cellular metabolic demands *in vivo*? Please provide evidence of this particularly considering the authors statement that vascularization requires 14 days. Please clarify

We thank the reviewer for highlighting this important point. As we demonstrated ecO_2 *in vitro* over time, we found that oxygenation in hypoxic cell cultivation can give rise to significant differences in terms of the viability even in a relatively shorter time scale. The cell viability of oxygenated cells was significantly greater than the non-oxygenated cells after 3 days ($p < 0.001$), and the fate of the cells was more obviously diverged after 10 days (**Figure 3c** in the main text or **Response Figure 6**; ***: $p < 0.001$, ****: $p < 0.0001$). Thus, clearly indicating that ecO_2 is providing the needed cell metabolic demand. Based on our findings *in vitro*, we implanted ecO_2 devices for 10 days which was the effective period to allow us to observe the differences in the cellular activities caused by hypoxia. The application of ecO_2 is a proof of concept of electrocatalytic oxygen production to sustain cells both *in vivo* and *in vitro*. Future application of the ecO_2 will explore its use as a transient O_2 supply toward vascularization of implanted cells or engineered tissue. This will necessitate development of different ecO_2 geometries and arrangements which are out on the scope on the current work.

Response Figure 6. Cell viability under hypoxic incubation as a function of time; blue – hypoxic incubation with ecO_2 ; red – hypoxic incubation without ecO_2 ; ***: $p < 0.001$, ****: $p < 0.0001$.

We have modified the manuscript in text ecO_2 supports implanted cells *in vivo* section accordingly-

"We demonstrated oxygen delivery to implanted cells in a rodent model (for additional information see Materials and Methods) for 10 days, as we observed that 10-day hypoxic incubation gave rise to the significant difference in cell viability."

7. Nanostructure of SIROF was examined using SEM before or post duty cycle. The images of SEM in high magnification were blurry. Please add the methods of SEM and duty cycles.

We thank the reviewer for pointing out that challenge. SIROF is known to have nanoscopic dendritic growth at the conditions we used (**Reference 45** in the main text²⁶). To clearly observe morphological changes such as nanoscopic features contributing to high-surface-area, we employed the secondary electron detector at an accelerating voltage of 1 kV which provides information mainly from surface features. The images were collected before and after 21-day *in vitro* oxygenation with 100 % duty cycle. The post-experiment images (**Supplementary Figure 7b**) showed degraded nanostructure in comparison to the pre-experiment samples (**Supplementary Figure 7a**). We revised our figure and added a few arrows to point out the features before (red arrows) and after (green arrows). Clearly, the dendritic features we see in panel **a** are degraded. Similar observations were documented in the literature by other research groups²⁵. (**Reference 45** in the Main text).

We have modified the manuscript in **Supplementary Note 3** and **Supplementary Figure 7** accordingly-

"All images were acquired with high-resolution (2048 x 1768 pixels) at an accelerating voltage of 1 kV with a working distance of 5 mm using secondary electron detector to investigate surface nanoscopic features."

Supplementary Figure 7. ecO_2 before and after 21-day oxygen evolution reaction. Representative SEM images (a) before and (b) after electrochemical oxygen evolution for 21 days with 100 % duty cycle load. The arrows indicated degraded dendritic structures (red: before, green: after); scale bars: i – 10 μm ; ii – 2 μm ; iii – 1 μm ; iv – 500 nm. All images were collected at an accelerating voltage of 1 kV with a working distance of 5 mm.

- For the LSV scan, the 1.6 V of potential voltage was used in 3-electrode setup while 2-electrode setups used 1.7 V. Can the authors add the clarification for this? Also, can the authors indicate the applied voltage used for the subsequent in vivo and in vitro experiments?

We thank the reviewer for this comment. 3-electrode setup which uses working electrode (catalyst), counter electrode and reference electrode provides standardized electrochemical characteristics, since the redox features are collected versus reference electrode that has fixed potential. Therefore, we utilized 3-electrode setup for electrochemical characterization to have standardized analytical results of catalysts (platinum and SIROF) to the community (**Figure 2a** in the main text). However, since our ecO_2 devices operate in 2-electrodes arrangement which is easier to fabricate and control, they do not have similar characteristics as the 3-electrode setup mainly because there is no real reference electrode as in 3-electrode setup. As shown in **Response Figure 8-1**, LSV scans in 2-electrode and 3-electrode setups are neither identical nor comparable, because potentiometric scan is performed against different potential standards. To elaborate on the discrepancies, LSV in anodic direction using 3-electrode setup scans from lower to higher potential versus reference electrode whose potential is not fluctuated. Therefore, 3-electrode setup is an appropriate setting for specifically materials serving working electrode. However, potential in 2-electrode setup is the electric field between anode and cathode, namely compliance, and it does not imply potential load on working electrode. Rather, 2-electrode setup is generally relevant to studies about a whole electrochemical cell such as batteries, fuel cells and electrocatalytic devices. For example, LSV scans using 3-electrode setup presented the activation signal of Ir (IV) and oxygen evolution onset around 1.6 V vs. RHE, showing

the electrochemical characteristics of SIROF. On the other hand, LSV scans in 2-electrode setup showed different behaviors including even broader activation shoulder (ca. 0.8 V wide) and oxygen evolution onset at around 1.55 V. Consequently, 2-electrode measurements provide better insights for practical applications of ecO₂ device. Based on our observation of onset drift (**Supplementary Figure 5**) and sufficient oxygen production (**Figure 2c**), the compliance of 1.7 V was selected to execute practical analysis such as oxygen and byproduct measurements, in vivo and in vitro experiment. We revised our figures and added potential profiles to **Supplementary Figure 20 and 28** to show the applied potential for *in vitro* and *in vivo* oxygenation, respectively. Note that current spikes other than potential adjustment *in vitro* oxygenation are due to periodical media exchange, while all spikes in 10-day *in vivo* oxygenation are due to potential increase.

Figure R8-1. LSV curves of SIROF. LSV scan in the anodic direction (a) using 2-electrode setup with the on-chip counter electrode of ecO₂ and (b) using 3-electrode setup with Pt mesh (counter electrode) and 1 M Ag/AgCl (reference electrode); potential in 3-electrode setup was converted to reversible hydrogen electrode (RHE). Data is presented with mean±SD ($n=4-8$).

We have modified the **Supplementary Figure 20** and **28** accordingly-

Supplementary Figure 20. Current profiles of *in vitro* oxygenation. Current profiles from (a) 3-day, (b) 10-day and (c) 21-day *in vitro* oxygenation; each color corresponding to a device. Note that spikes in the current profiles from 21-day *in vitro* oxygenation other than potential adjustment are due to media exchange; for example, green arrows indicate current spikes caused by media exchange with deoxygenated media

Supplementary Figure 28. Current profiles of *in vivo* oxygenation. (a-d) Current profiles from each implanted ecO₂ device. All spikes in (b) are corresponding to potential increase, because media exchange was unavailable for the implanted devices. Note that (a) has 7 arrays due to the array failure.

9. For how long was the media de-oxygenated? Was there any FEA based analysis to approximate the de-oxygenation or was it based on measurements? Low oxygen levels should be below 2%. Please clarify the reason of why you chosen 1% oxygen instead of 2% or completely removing the oxygen (0% oxygen) in the manuscript and elaborate how to prepare the deoxygenated media.

We would like to address this comment in two parts as below.

1. De-oxygenation of media

We deoxygenated the media before in vitro experiments by transferring it into a ventilated T150 flask and placing it in an oxygen-controlled incubator (1% O₂) for at least 24 hours. This is also reported elsewhere in the literature for deoxygenating culture media²⁷. Natural gas diffusion allowed gas exchange between the media and the ambient atmosphere of the incubator, gradually deoxygenating the media. We also conducted an additional experiment (**Response Figure 9**) in which an oxygen sensor was placed in a 1% oxygen incubator. After five minutes, the sensor was immersed in the equilibrated media (400µl, DMEM:F-12) placed inside a PDMS well. Aside from a small spike in oxygen concentration caused by opening the incubator door, the media equilibrated back to the deoxygenated state within approximately 10 minutes.

2. Choice of deoxygenation levels

The choice of 1% hypoxia is highly relevant within the scope of our study. Oxygen sensing system in the cells makes the absolute pO₂ level in the microenvironment of cells irrelevant²⁸. Mitochondrial pO₂ levels could be as low as 0.061% which is 10-100 times lower than intracellular pO₂²⁹. Several organotypic and cell type-specific characteristics could result in a huge variety of pO₂ levels. Tissues such as medulla, renal artery, efferent arterioles, and kidney function physiologically at oxygen levels close to 1%³⁰. Other tissues such as avascular cornea, and some cell types such as neurons strikingly reside in low pO₂. Some cells such as human multipotent mesenchymal stromal cells³¹ are adversely affected by 1% hypoxia, and result in reduction in its proliferation and differentiation. Thus, the selection of 1% hypoxia is well-founded, and aligns with physiologically relevant pO₂ levels reported in the literature. Therefore, for this study, we conducted all the in vitro experiments at the minimum attainable oxygen level in the incubator, which is 1% pO₂.

Response Figure 9. deoxygenation of the equilibrated media

10. Authors have cultured cells at normal oxygen level and found that cell death, indicating hypoxic death. Is it possible that the cell death occurred due to the pre-existing cell death or the effect of the alginate on cells? Were the cells characterized or cell viability assayed before mixing with the alginate? Please add those results as well as discuss the effect of alginate on cell viability?

We thank the reviewer for the suggestion. We would like to add a bit more information for the work with the alginate capsules. Alginate capsules were developed in Dr. Veisoh lab and have been demonstrated to be highly stable (Reference 46 in the main text³²). Obtained capsules are maintained under incubation for 24 hours before transfer to the ecO₂ devices. Prior to loading the ecO₂ with alginate capsules we do check the viability of the capsules to make sure that the starting viability is > 90 %. This analysis is noted in our MS figures as Day 0 (Figure 3c in the main text). We note that the Day 0 viability analysis is performed either *in vitro* or *in vivo* (with or without oxygenation). Please find Figure 3c which shows viability at Day 0.

Response Figure 10. A representative set of Live/dead assay fluorescence images from live/dead assay of the capsules at Day 0. (a) Day 0 of *in vitro* oxygenation and (b) Day 0 of *in vivo* oxygenation; blue – Hoechst (nuclei); green – calceinAM (live); red – ethidium homodimer (dead); all scale bars are 200 μm

We have modified the manuscript in text the Materials and Methods section accordingly -

"Note that Day 0 live/dead assay was performed prior to all long-term cultivation of capsules to exclude the influence on the viability other than incubation conditions."

11. The authors indicated that there was no significant inflammation and foreign body response post in vivo study. Is there any data of immune cells in support this statement?

We thank the reviewer for pointing this out. The immune response comment is based on the explant day where we did not see fibrotic tissue forming around the implant based on visual examination (See **Response Figure 11, Supplementary Figure 23**). Future work will include thorough studies of the responses to application specific devices. We have modified the text (and reduced our prior claims) to address this point.

We modified the manuscript in text ecO₂ supports implanted cell in vivo section accordingly-

"Based on the visual inspection upon explantation of the devices, there was no apparent indication of fibrosis or fluid accumulation from inflammation on either device or the implanting locations (**Supplementary Figure 23**)"

Response Figure 11. ecO₂ post 10 days *in vivo*. A representative image of retrieved ecO₂ after 10-day *in vivo*. scale bar – 1 cm.

12. The authored reported that the varied cell viability between in vitro and in vivo was due to the different volume of media per cell. Please add the media for in vivo study in methods and clarify whether the media were used and changed in the vivo study.

We thank the reviewer for highlighting this point. We used the same media in both in vitro and in vivo cell oxygenation, which was formulated with DMEM:F12, 10 % fetal bovine serum and 1 % antibiotics. We note that media was not exchanged in animal studies because of the inaccessibility of the implanted device.

We have modified the manuscript in text Supplementary Note section accordingly-

"Note that the media exchange was unable to do due to inaccessibility to the device during implantation."

Minor comments:

1. Typo on Supplementary Fig 24 & 25 'what are the colors' at the caption has been left from the corrections.

We have corrected the figure captions with the revised figure number.

"**Supplementary Figure 27. z-stack analysis of 10-day in vivo control;** (a) 3D-reconstructed z-stacked images and the location of each presented image. (b) z=100 μ m; (c) z=0 μ m; (d) z=-100 μ m. Scale bars are 100 μ m. Green: Calcein (Live), Red: B (Dead), Blue: C (Nuclei)."

"**Supplementary Figure 28. Current profiles of in vitro oxygenation.** Current profiles from (a) 3-day, (b) 10-day and (c) 21-day in vitro oxygenation; each color corresponding to a device. Note that spikes in the current profiles from 21-day in vitro oxygenation other than potential adjustment are

due to media exchange; for example, green arrows indicate current spikes caused by media exchange.”

2. Dense cells were encapsulated into chips as PDMS enclosure applied. There was no reaction or integration with host tissue, which limits the implications of in vivo applications. Please address and clarification this assumption for the in vivo model.

We thank the reviewer for commenting on this point. We designed the implantable ecO₂ with an emphasis to provide selective mass transport and nutrients other than oxygen to the encapsulated cells and minimize inflammation caused to the host animal. The therapeutic cells were isolated from the host through two mechanisms. First, the alginate hydrogels encapsulating the cells, second, a nano-porous size-selective membrane (pore size ~400 nm) that was installed on top of the cell chamber. For minimizing the inflammation, we used PDMS for enclosing all the implants, and the cell chamber was applied using a medical-grade epoxy onto the oxygenator chips. This allowed mass transport for smaller components, for example, nutrients and metabolic wastes through the membrane, while limiting immune responses from the host. The pore size of the membrane was selected based on previous studies in immunoisolation device which has selective permeability to screen large components causing implantation failure^{16,33}. For this concept, it is beneficial that fibrotic formation or other foreign body reactions do not occur on the device, while mass transport to support the implanted cells with nutrients other than oxygen is facilitated³¹. We appreciate the reviewer’s comment in our in vivo model, and please see our description for the model we used in the “ecO₂ supports implanted cells in vivo” section-

“While medical grade PDMS served as impermeable housing for electronics, a microporous polycarbonate membrane was added at the top of the cell compartment to provide selective mass transport and nutrients other than oxygen (Supplementary Figure 21 and 23)⁸.”

3. The safety and accessibility of this system plays the important role in transplanted cell-based-therapies. A battery-free wireless power can make the system more flexible rather than battery power and connecting cables. Please comment on this.

We would like to emphasize that the battery was used in the current iteration of ecO₂ to primarily demonstrate the technology in vivo, and for simplicity. We anticipate our approach is not limited to the current setup and can benefit from various existing and emerging technologies in wireless power transfer and communication. We already commented on this in the Conclusion section *“For example, while we use batteries in the implant in this work to ease experimentation, the low power requirements (ca. 1.25 μW) (especially at low duty cycling) can be readily integrated with wireless power/communications technologies with more favorable safety profiles (i.e., radio frequency/ultrasound/magnetolectric^{14,18}).”*

4. For the in vivo, the channels have not been compartmentalized. Is it possible to enlarge the size of arrays for producing large amounts of oxygen? Please comment on this in future direction.

The number of devices in ecO₂ can be varied depending on the specific requirements of the therapy, such as the desired cell volume or total cell number. We would like to point to our response to Q1 of reviewer 3 for more details.

We have added the following to the main text in the Conclusion section

“Also, the number of devices in ecO₂ can be varied depending on the specific requirements of the therapy, such as the desired cell volume, total cell number, or needed oxygenated area.”

Reviewer #4 (Remarks to the Author):

I co-reviewed this manuscript with one of the reviewers who provided the listed reports as part of the Nature Communications initiative to facilitate training in peer review and appropriate recognition for co-reviewers.

We thank the reviewer for their comment. We have addressed the comments in our response above.

References

1. Foller, P. C. & Tobias, C. W. The Anodic Evolution of Ozone. *J Electrochem Soc* **129**, 506–515 (1982).
2. Lyle, H., Singh, S., Paolino, M., Vinogradov, I. & Cuk, T. The electron-transfer intermediates of the oxygen evolution reaction (OER) as polarons by *in situ* spectroscopy. *Physical Chemistry Chemical Physics* **23**, 24984–25002 (2021).
3. Li, Y. *et al.* Direct Electrochemical Measurements of Reactive Oxygen and Nitrogen Species in Nontransformed and Metastatic Human Breast Cells. *J Am Chem Soc* **139**, 13055–13062 (2017).
4. Rahimi, E. *et al.* Role of phosphate, calcium species and hydrogen peroxide on albumin protein adsorption on surface oxide of Ti6Al4V alloy. *Materialia (Oxf)* **15**, 100988 (2021).
5. Kawasaki, K. & Kamagata, Y. Phosphate-Catalyzed Hydrogen Peroxide Formation from Agar, Gellan, and κ -Carrageenan and Recovery of Microbial Cultivability via Catalase and Pyruvate. *Appl Environ Microbiol* **83**, (2017).
6. Zhao, Y. *et al.* Janus electrocatalytic flow-through membrane enables highly selective singlet oxygen production. *Nat Commun* **11**, 6228 (2020).
7. Chen, G. *et al.* A discussion on the possible involvement of singlet oxygen in oxygen electrocatalysis. *Journal of Physics: Energy* **3**, 031004 (2021).
8. Codorniu-Hernández, E. & Kusalik, P. G. Mobility Mechanism of Hydroxyl Radicals in Aqueous Solution via Hydrogen Transfer. *J Am Chem Soc* **134**, 532–538 (2012).
9. Zhao, X., Liu, X., Huang, B., Wang, P. & Pei, Y. Hydroxyl group modification improves the electrocatalytic ORR and OER activity of graphene supported single and bi-metal atomic catalysts (Ni, Co, and Fe). *J Mater Chem A Mater* **7**, 24583–24593 (2019).
10. Pang, Y., Xie, H., Sun, Y., Titirici, M. M. & Chai, G. L. Electrochemical oxygen reduction for H₂O₂ production: Catalysts, pH effects and mechanisms. *Journal of Materials Chemistry A* vol. 8 24996–25016 Preprint at <https://doi.org/10.1039/d0ta09122g> (2020).
11. Naito, T., Shinagawa, T., Nishimoto, T. & Takanabe, K. Recent advances in understanding oxygen evolution reaction mechanisms over iridium oxide. *Inorganic Chemistry Frontiers* vol. 8 2900–2917 Preprint at <https://doi.org/10.1039/d0qi01465f> (2021).
12. Bianchi, G., Mazza, F. & Mussini, T. Catalytic decomposition of acid hydrogen peroxide solutions on platinum, iridium, palladium and gold surfaces. *Electrochim Acta* **7**, 457–473 (1962).
13. Bashor, C. J., Hilton, I. B., Bandukwala, H., Smith, D. M. & Veisoh, O. Engineering the next generation of cell-based therapeutics. *Nat Rev Drug Discov* **21**, 655–675 (2022).

14. Chen, J. C. *et al.* A wireless millimetric magnetoelectric implant for the endovascular stimulation of peripheral nerves. *Nat Biomed Eng* **6**, 706–716 (2022).
15. Jiang, D. *et al.* Emerging Implantable Energy Harvesters and Self-Powered Implantable Medical Electronics. *ACS Nano* **14**, 6436–6448 (2020).
16. Ludwig, B. *et al.* Transplantation of human islets without immunosuppression. *Proceedings of the National Academy of Sciences* **110**, 19054–19058 (2013).
17. Maharbiz, M. M., Holtz, W. J., Sharifzadeh, S., Keasling, J. D. & Howe, R. T. A microfabricated electrochemical oxygen generator for high-density cell culture arrays. *Journal of Microelectromechanical Systems* **12**, 590–599 (2003).
18. Maleki, T. *et al.* An Ultrasonically Powered Implantable Micro-Oxygen Generator (IMOG). *IEEE Trans Biomed Eng* **58**, 3104–3111 (2011).
19. Liang, J.-P. *et al.* Engineering a macroporous oxygen-generating scaffold for enhancing islet cell transplantation within an extrahepatic site. *Acta Biomater* **130**, 268–280 (2021).
20. Wang, L.-H. *et al.* An inverse-breathing encapsulation system for cell delivery. *Sci Adv* **7**, 5835–5849 (2021).
21. Desai, T. & Shea, L. D. Advances in islet encapsulation technologies. *Nat Rev Drug Discov* **16**, 338–350 (2017).
22. Schukur, L., Geering, B., Charpin-El Hamri, G. & Fussenegger, M. Implantable synthetic cytokine converter cells with AND-gate logic treat experimental psoriasis. *Sci Transl Med* **7**, (2015).
23. Carmeliet, P. & Jain, R. K. Angiogenesis in cancer and other diseases. *Nature* **407**, 249–257 (2000).
24. Kurosawa, H., Matsumura, M. & Tanaka, H. Oxygen diffusivity in gel beads containing viable cells. *Biotechnol Bioeng* **34**, 926–932 (1989).
25. Lovett, M., Lee, K., Edwards, A. & Kaplan, D. L. Vascularization Strategies for Tissue Engineering. *Tissue Eng Part B Rev* **15**, 353–370 (2009).
26. Negi, S., Bhandari, R., Rieth, L., Van Wagenen, R. & Solzbacher, F. Neural electrode degradation from continuous electrical stimulation: Comparison of sputtered and activated iridium oxide. *J Neurosci Methods* **186**, 8–17 (2010).
27. Wenger, R., Kurtcuoglu, V., Scholz, C., Marti, H. & Hoogewijs, D. Frequently asked questions in hypoxia research. *Hypoxia* **35** (2015) doi:10.2147/HP.S92198.
28. Stiehl, D. P. *et al.* Increased Prolyl 4-Hydroxylase Domain Proteins Compensate for Decreased Oxygen Levels. *Journal of Biological Chemistry* **281**, 23482–23491 (2006).

29. Larsen, F. J. et al. Mitochondrial oxygen affinity increases after sprint interval training and is related to the improvement in peak oxygen uptake. *Acta Physiologica* 229, (2020).
30. Keeley, T. P. & Mann, G. E. Defining Physiological Normoxia for Improved Translation of Cell Physiology to Animal Models and Humans. *Physiol Rev* 99, 161–234 (2019).
31. Holzwarth, C. et al. Low physiologic oxygen tensions reduce proliferation and differentiation of human multipotent mesenchymal stromal cells. *BMC Cell Biol* 11, 11 (2010).
32. Bochenek, M. A. *et al.* Alginate encapsulation as long-term immune protection of allogeneic pancreatic islet cells transplanted into the omental bursa of macaques. *Nat Biomed Eng* 2, 810–821 (2018).
33. Bose, S. et al. A retrievable implant for the long-term encapsulation and survival of therapeutic xenogeneic cells. *Nat Biomed Eng* 4, 814–826 (2020).

REVIEWERS' COMMENTS

Reviewer #3 (Remarks to the Author):

The authors put a good effort into replying to the comments made by myself and the other reviewers. I believe they have addressed all my comments.

Reviewer #4 (Remarks to the Author):

I co-reviewed this manuscript with one of the reviewers who provided the listed reports.